# Large-sample assessment of varying spatial resolution on the streamflow estimates of the wflow_sbm hydrological model

Jerom P.M. Aerts[1], Rolf W. Hut[1], Nick C. van de Giesen[1], Niels Drost[2], Willem J. van Verseveld[3], Albrecht H. Weerts[4,5], and Pieter Hazenberg[6]

[1]Water Resources Section, Faculty of Civil Engineering and Geosciences, Delft University of Technology, Stevinweg 1, 2628 CN Delft, the Netherlands
[2]Netherlands eScience Center, Science Park 140, 1098 XG Amsterdam, the Netherlands
[3]Catchment and Urban Hydrology, Department of Inland Water Systems, Deltares, P.O. Box 177, 2600MH Delft, The Netherlands
[4]Operational Water Management, Department of Inland Water Systems, Deltares, P.O. Box 177, 2600MH Delft, The Netherlands
[5]Hydrology and Quantitative Water Management Group, Wageningen University and Research, P.O. Box 47, 6700AA Wageningen, The Netherlands
[6]Applied Research Center, Florida International University, FL 33174, Miami, the United States of America.

**Correspondence:** Jerom Aerts (J.P.M.Aerts@tudelft.nl)

**Abstract.** Distributed hydrological modelling moves into the realm of hyper-resolution modelling. This results in a plethora of scaling related challenges that remain unsolved. In light of model result interpretation, finer resolution output might implicate to the user an increase in understanding of the complex interplay of heterogeneity within the hydrological system. Here we investigate spatial scaling in the form of varying spatial resolution by evaluating the streamflow estimates of the distributed wflow_sbm hydrological model based on 454 basins from the large-sample CAMELS data set. Model instances are derived at 3 spatial resolutions, namely 3 km, 1km, and 200m. The results show that a finer spatial resolution does not necessarily lead to better streamflow estimates at the basin outlet. Statistical testing of the objective function distributions (KGE score) of the 3 model instances resulted in only a statistical difference between the 3km and 200m streamflow estimates. However, an assessment of sampling uncertainty shows high uncertainties surrounding the KGE score throughout the domain. This makes the conclusion based on the statistical testing inconclusive. The results do indicate strong locality in the differences between model instances expressed by differences in KGE scores of on average 0.22 with values larger than 0.5. The results of this study open up research paths that can investigate the changes in flux and state partitioning due to spatial scaling. This will help further understand the challenges that need to be resolved for hyper resolution hydrological modelling.

## 1 Introduction

Hydrological model development follows competing model philosophies (Hrachowitz and Clark, 2017). From one end of the spectrum to the other these include high resolution small-scale process resolving distributed models and spatially lumped conceptual models. All model structures developed within the competing philosophies have their own limitations and advantages.

There are overlapping challenges that all models face among which parameter estimation and the representation of spatial heterogeneity (Clark et al., 2017).

The parameter identifiability problem stems from the inability to obtain unique and realistic parameters at the modelling scale due to structural model deficiencies and applied calibration techniques (Sorooshian and Gupta, 1983). Multiple studies have extensively researched calibration techniques to overcome the parameter identifiability problem (e.g. Sorooshian and Gupta, 1983; Vrugt et al., 2002; Guse et al., 2020). The identifiability problem is emphasized in distributed modelling, the focus of this study, by the limitation of parameter measurements not being compatible with the modelling scale (Grayson et al., 1992). This

results in the need for transferring parameters in space and time. Multiple studies have looked into parameter transferability (e.g. Finnerty et al., 1997; Haddeland et al., 2006; Wagener and Wheater, 2006). Melsen et al. (2016) discussed that the inadequacy of transferring parameters in space and time may indicate a lack in spatial heterogeneity and temporal variability representation in the models. Methods such as the multi-scale parameter regionalization technique (MPR) Samaniego et al. (2010) emerged to increase the transferability of parameters from the data resolution to the hydrological model resolution.

Imhoff et al. (2020) used a different method that applied pedo-transfer functions to derive parameters at the finest available data resolution and upscale to the various spatial model resolutions of the wflow_sbm hydrological model.

    The effects of spatial heterogeneity has been studied at a catchment scale using the representative elementary watershed (REW) theory developed by (Wood et al., 1988; Reggiani et al., 1998, 1999; Reggiani and Schellekens, 2003). In Wood et al. (1988) the basin was divided in sub-basins and then aggregated to basin level to study scaling behavior. This type of research

is still very relevant as in recent years, the hydrologic modelling community has surpassed the REW scale threshold of 1 km2 with the move towards so called hyper-resolution modelling. The discussion following this move revealed that in addition to the many benefits, e.g. applicability for stakeholders, there are multiple challenges to address (Wood et al., 2011; Beven and Cloke, 2012; Bierkens et al., 2015). These challenges include scaling issues (Gupta et al., 1986; Blöschl and Sivapalan, 1995) such as; (1) the need to explicitly model processes that are parameterized at coarser resolutions, (2) lateral connections

between compartments of the hydrological system that are averaged out or ignored at coarser resolutions, and (3) an increase in uncertainty due to lacking process and parameter knowledge due to insufficient data quality at finer resolutions (Bierkens et al., 2015).

    The scaling issues arise when the (often unconscious) assumption is made that a hydrological model used at various spatial and temporal resolutions should estimate similar states and fluxes independent of scale. An Utopian model has scale-invariant

model parameterization and hydrological process descriptions. The development of scale-invariant hydrological models is, however, very challenging as most hydrological processes do not scale in a linear manner (e.g. Bras, 2015; Rouholahnejad Freund et al., 2020). Instead processes at one length scale influence those at other scales (Horritt and Bates, 2001).

    Because of the complex nature of scaling issues and a shifting distributed modelling climate towards hyper-resolution modelling it is important to continuously assess the effects of scaling. Without investigating what this move entails the hydrological

modelling community risks communication problems with the users of model results. In the case of spatial model resolution, the increase in level of detail in model output might implicate to the user an increase in understanding of the complex interplay

of heterogeneity within the hydrological system. We can only determine this by continuously assessing how models behave under various spatial (and temporal) resolutions.

Multiple studies have looked into spatial scaling effects by varying spatial model resolution. Booij (2005) found that increasing the spatial resolution of a semi-lumped HBV hydrological model only marginally increased model performance based on streamflow estimates. The coupled ParFlow-CLM model was evaluated with various grid cell sizes by Shrestha et al. (2015) and they found, among others effects, that soil moisture estimates were grid cell size dependent. Sutanudjaja et al. (2018) introduced the transition from 30 arc minutes to 5 arc minutes grid cell size simulations of the distributed PCR-GLOBWB model. Results showed a general increase in model performance compared to streamflow observations. However, regional scaling issues were present. In some of the basins model performance was lower at a finer spatial model resolution. This study made it apparent that a large sample of hydrological diverse basins should be considered when investigating the effects of varying spatial resolution. To our knowledge there are no studies that have looked into varying spatial resolution within the hyper-resolution realm on a large-sample of basins. This will be the focus of this research.

The distributed conceptual wflow simple bucket model (wflow_sbm) (Schellekens et al., 2020) utilizes high resolution datasets to derive model instances globally at varying spatial resolution. Parameter estimates are based on the work of Imhoff et al. (2020) to ensure consistency across scale. Remotely sensed soil and land cover data sets are sources for estimating parameters through pedo-transfer functions (PTF) (e.g. Brakensiek et al., 1984; Cosby et al., 1984). PTFs are a collection of predictive functions, so called super parameters (Tonkin and Doherty, 2005), derived at point-scale that estimate soil parameters where underlying data is scarce. For most wflow_sbm model parameters a priori parameters are available. This is not (yet) the case for the horizontal conductivity fraction (KsatHorFrac) parameter, making it a logical parameter for calibration as it is also one of the more sensitive parameters in the model (Imhoff et al., 2020). The flexible setup of wflow_sbm can be used to assess spatial scaling issues due to quasi-scale invariant parameters whilst maintaining similar hydrological process descriptions across scales. This setup includes the recent improvements by Eilander et al. (2021) who developed a scale-invariant method for upscaling river networks (one of the suggested causes of the inconsistent streamflow estimates across scales as shown in Imhoff et al. (2020).

In this study we quantify the effects of varying spatial resolution on the wflow_sbm streamflow estimates for a large-sample of hydrological diverse basins in the CAMELS dataset (Newman et al., 2015; Addor et al., 2017). By conducting this research on a large-sample of basins we can assess the results on consistency and locality. The assessment is conducted by creating 3 model instances at varying spatial resolutions for each basin: a 3km, 1km, and 200m spatial grid resolution. These instances cover a broad range of large and small scale dynamics. For example: snow accumulation at the mountain range (> 1km) and the mountain ridge scale (< 1km) (e.g. Houze Jr., 2012; Mott et al., 2018; Vionnet et al., 2021), or closing in on the hillslope scale (< 100m) (e.g. Tromp-van Meerveld and McDonnell, 2006; Fan et al., 2019). The parameters for the wflow_sbm model instances are estimated at the highest available data resolution and aggregated to the modelling grid using the upscale rules as defined in Imhoff et al. (2020).

Our hypothesis is that the differences in streamflow estimates at various spatial resolutions will be small due to the parameters being quasi-scale invariant and hydrological process descriptions in the model remaining the same across spatial scales. We

will reject this hypothesis when the results show significantly different streamflow estimates across the studied resolutions. In addition with this study, we showcase how the eWaterCycle platform (Hut et al., 2021) can be utilized for computational intensive large-sample modelling studies.

## 2 Methodology

### 2.1 Input data

#### 2.1.1 The CAMELS data set

The CAMELS data set is a collection of hydrologically relevant data on 671 basins located in the Contiguous United States (CONUS) (Newman et al., 2015). The basins were selected based on a minimum amount of human influence on the hydrological system, e.g. the absence of large reservoirs. The data set includes 20 years of continuous streamflow records from 1990 to 2009 from the United States Geological Survey (USGS). The CAMELS data set covers a hydrological and climatological diverse selection of basins. The sample-size, hydrological diversity of basins, and common use of this data set in other hydrological modelling studies (e.g. Knoben et al., 2020; Gauch et al., 2021) are the reasons for selecting this case study area.

Of the 671 basins we ran 567 basins successfully for each of the 3 model instances (i.e. 3km, 1 km and 200m resolution). Reasons for excluding basins in our analyses are missing streamflow observations (7 basins) or errors during parameter estimation (97 basins). Parameter estimation errors occurred mainly during drainage network delineation, either when the basin outlet consisted of a single grid cell that results in a model coding error or when inconsistencies occurred in the local drainage direction layer. When a single model instance of the 3 model instances failed the basin was excluded from further analyses. Figure 1 shows the locations of the included and excluded basins as well as the reason for exclusion.

#### 2.1.2 Streamflow Observations

The United States Geological Survey (USGS) streamflow observation records were downloaded to match our model simulation period from 1996 to 2016. The data is resampled to daily data and the units were converted to m3*s-1. We ensured consistency in time zones between the observation and the model simulations by matching the USGS streamflow data with the UTC time zone. The tooling used for downloading, resampling, unit conversion, and shifting of time zones might be of interest to others in the hydrological community and is available in the GitHub repository (https://github.com/jeromaerts/eWaterCycle_example_notebooks).

#### 2.1.3 Meteorological input and pre-processing

The meteorological input requirements of the wflow_sbm model are precipitation, temperature and potential evapotranspiration. Precipitation data was obtained from the Multi-Source Weighted-Ensemble Precipitation Version 2.1 (MSWEP) (Beck et al., 2019). The data set was constructed using bias corrected gauge, satellite and reanalyses data. The data is available at 0.1 degrees spatial (11 km) and 3-hourly temporal resolution for the period 1979 to 2017. The temperature variable was obtained from the ERA5 reanalyses data set (Hersbach et al., 2020). The data is available at 0.25 degrees (31 km) spatial and a 1-hourly

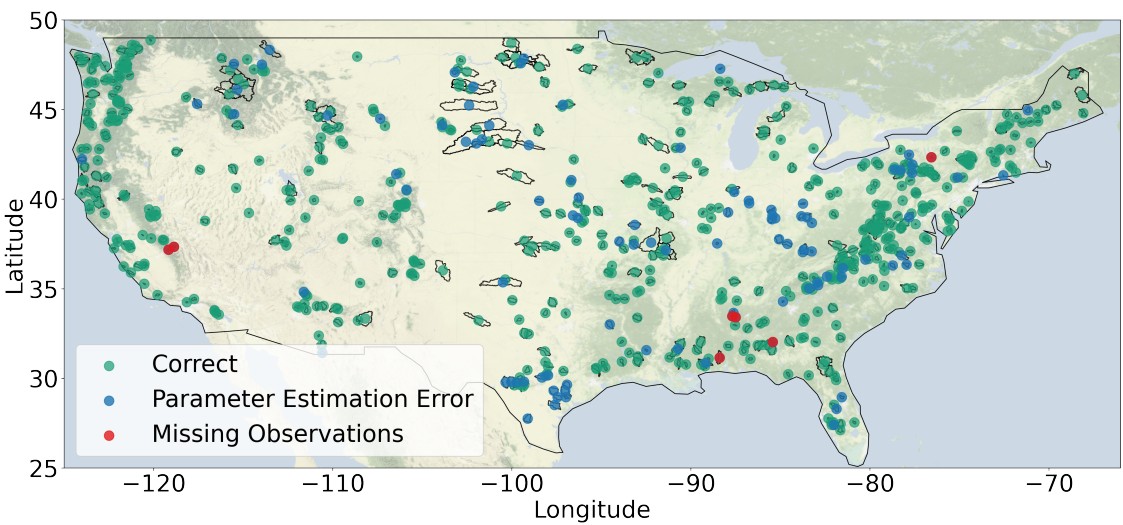

**Figure 1.** Basin locations of the CAMELS data set with in green the included basins. In blue excluded basins due to parameter estimation errors and in red excluded basins due to missing streamflow observations. Basemap made with Natural Earth.

temporal resolution. In addition to temperature, we used ERA5 variables to calculate potential evapotranspiration using the De Bruin method (Bruin et al., 2016).

We conducted a preliminary analysis for 6 basins were we compared model simulations based on streamflow estimates that use ERA5 precipitation to those that use MSWEP precipitation. Results indicated that use of ERA5 precipitation did not produce desirable streamflow estimates compared to MSWEP precipitation. Switching to the MSWEP precipitation product improved streamflow estimates throughout the case study area. Figure A1 and Table A1 in the Appendix A contain the results of this analysis. Noticeable is that some of the basins are very sensitive to changes due to different forcing data sets as shown by the streamflow based objective functions.

The meteorological input is pre-processed within the eWaterCycle platform using the Earth System Model Evaluation Tool (ESMValTool) V2.0 (Righi et al., 2020; Weigel et al., 2021). Before further processing the data is aggregated to daily values. The precipitation variable is disaggregated to the modelling grid using the second-order conservative method to ensure consistency of the total volume of precipitation across spatial scales. The temperature variable is disaggregated with the en-

**Table 1.** Overview of data sources for parameter estimation with categories, references, and version.

| Data set | Category | Reference | Version |
|----------|----------|-----------|---------|
| Merit Hydro | topography | Yamazaki et al. (2019) | 1.0 |
| GRAND (hydro_reservoirs) | surface water | Lehner et al. (2011) | 1.0 |
| hydroLAKES (hydro_lakes) | surface water | Messager et al. (2016) | 1.0 |
| Randolph Glacier Inverntory | surface water | Pfeffer et al. (2014) | 6.0 |
| CHELSA | meteo | Karger et al. (2017) | 1.2 |
| Köppen-Geiger | meteo | Kottek et al. (2006) | 2017 |
| VITO | landuse & landcover | Buchhorn et al. (2020) | v2.0.2 |
| Modis LAI | landuse & landcover | Myneni et al. (2015) | MCD15A3H V006 |
| SoilGrids | soil | Hengl et al. (2017) | 2017 |

vironmental lapse rate and the Digital Elevation Model (DEM) used by the hydrological model. The variables required by the De Bruin method (Bruin et al., 2016) are disaggregated using the (bi)linear method and subsequently used to calculate potential evapotranspiration. The code for used for pre-processing is included in the Jupyter Notebooks made available with this manuscript (DOI:10.5281/zenodo.5724512).

### 2.1.4 Parameter estimation from external data sources

The parameter sets used in this study were derived using the hydroMT software package (Eilander and Boisgontier, 2021). The data sources for deriving parameter sets are open-source global data sets. These include topography, surface water, landcover & landuse, soil, meteo, and river gauge data. The pedo-transfer function to estimate soil properties is based on Brakensiek et al. (1984). In 25 of the 567 basins, lakes and or reservoirs were included in the model parameters given a threshold area of 1 km2 and 10 km2 respectively.

An overview of the data and references are provided in Table 1.

## 2.2 Model Experiment Setup

### 2.2.1 The wflow_sbm model (v.2020.1.2)

The wflow_sbm model is available as part of the wflow open-source modeling framework (Schellekens et al., 2020), which is based on PCRaster (Karssenberg et al., 2010) and Python. Figure 2 shows the different processes and fluxes that are part of the wflow_sbm hydrological concept. The soil part of wflow_sbm model is largely based on the Topog_SBM model (Vertessy and Elsenbeer, 1999), that considers the soil as a "bucket" with a saturated and unsaturated store. This model was developed for simulating small-scale hydrology. For channel, overland and lateral subsurface flow a kinematic wave approach is used, similar to TOPKAPI (Benning, 1995; Ciarapica and Todini, 2002), G2G (Bell et al., 2007), 1K-DHM (Tanaka and Tachikawa, 2015) and Topog_SBM (Vertessy and Elsenbeer, 1999). Wflow_sbm has a simplified physical basis with parameters that represent

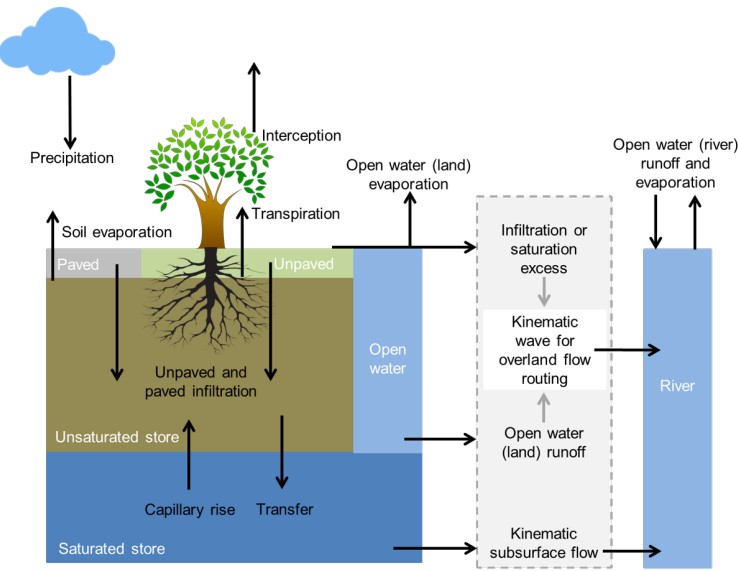

**Figure 2.** Overview of the different processes and fluxes in the wflow_sbm model (Schellekens et al., 2020).

physical characteristics, leading to (theoretically) an easy linkage of the parameters to actual physical properties. Topog_SBM
is mainly used to simulate fast runoff processes during discrete storm events in small basins (< 10 km2) (evapotranspiration
losses are ignored). Since evapotranspiration losses and capillary rise were added to wflow_sbm, the derived wflow_sbm
approach can be applied to a wider variety of basins. The main differences of wflow_sbm with Topog_SBM are:

– The addition of evapotranspiration and interception losses (Gash model (Gash (1979)) on daily time steps or a modified
  Rutter model on subdaily time steps (Rutter et al., 1971, 1975)).

– The addition of a root water uptake reduction function (Feddes et al., 1978).

– The addition of capillary rise.

– The addition of glacier and snow build-up and melting processes.

– Wflow_sbm routes water over an eight direction (D8) network, instead of the element network based on contour lines
  and trajectories, used by Topog_SBM.

– The option to divide the soil column into any number of different layers.

– Vertical transfer of water is controlled by the saturated hydraulic conductivity at the water table or bottom of a layer, the
  relative saturation of the layer, and a power coefficient depending on the soil texture (Brooks and Corey, 1964).

### 2.2.2 Model Runs & Calibration

We derived 3 model instances at varying spatial model resolution that cover a 3km, 1km, and 200m grid. While for most parameters of the wflow_sbm model a priori estimates can be derived from external sources, a single non-distributed parameter needs to be calibrated for each basin: the saturated horizontal conductivity often expressed as a fraction (KsatHorFrac) of the vertical conductivity. This parameter cannot be derived from external data sources because it compensates for anisotropy, unrepresentative point measurements of the saturated vertical conductivity, and model resolution (Schellekens et al., 2020). A sensitivity analyses conducted by the model developers concluded that the KsatHorFrac is the most effective parameter when it comes to calibration based on streamflow estimates, also briefly discussed in Imhoff et al. (2020). Increasing the value of this parameter leads to an increased base flow component and reduced peak flow and flashiness.

We calibrated the models to match model setups of those used by the users of the hydrological model. The model instances are calibrated using the modified Kling-Gupta Efficiency score (KGE) (Gupta et al., 2009; Kling et al., 2012) by comparing the simulated streamflow estimates with the streamflow observations at the basin outlet. 21 runs are evaluated based on an interval of KsatHorFrac values ranging between 1 and 1000. The best performing model run and its corresponding KsatHorFrac parameter are selected for further analyses during the evaluation period. The model calibration is conducted from 1997 to 2005 and the model evaluation from 2007 to 2016. The years 1996 and 2006 are regarded as spin-up years and are not included in the analyses.

### 2.3 Benchmark Selection

To select basins with model performance considered to be at least reasonably well, we applied a statistical benchmark to beat. The use of a benchmark allows for better interpretation of objective function based results (Garrick et al., 1978; Pappenberger et al., 2015; Schaefli and Gupta, 2007; Seibert, 2001; Seibert et al., 2018; Knoben et al., 2020). We adopt, in part, the same methodology for statistical benchmark creation as Knoben et al. (2020). The benchmark is created by calculating the mean and median of streamflow observations per calendar day of the 10 year evaluation period and the results are compared to streamflow predictions from the model. KGEs are calculated for the observed streamflow versus these mean and median values. The mean is expected to better represent larger basins with more stable flow regimes whilst the median better covers the flashiness of smaller headwater basins. This benchmark serves as a lower boundary for the model predictions: if the model, for any of the 3 resolutions, has a lower KGE than either the median or mean flow benchmark, it is considered not suited for this study and removed from further analyses.

### 2.4 Analyses of results

#### 2.4.1 Objective function and sampling uncertainty

We use the modified Kling-Gupta Efficiency metric for the analyses of results. Ideal model performance has a KGE score of 1 and a KGE score of -0.41 is equal to taking the mean flow as a benchmark (Knoben et al., 2019c). The KGE score is defined

by its 3 components, the Pearson correlation, the mean bias, and the variability bias. The lowest value of the 3 components determines the KGE score. The KGE score and its components:

$$KGE' = 1 - \sqrt{(r-1)^2 + (\beta - 1)^2 + (\gamma - 1)^2} \tag{1}$$

$$\beta = \left( \frac{\mu_s}{\mu_o} \right) \tag{2}$$

$$\gamma = \left( \frac{\sigma_s / \mu_s}{\sigma_o / \mu_o} \right) \tag{3}$$

,with $r$ the Pearson correlation, $\beta$ the mean bias, and $\gamma$ the variability bias. $\mu_s$ and $\sigma_s$ are the mean and standard deviation of the simulations. $\mu_o$ and $\sigma_o$ are the mean and standard deviation of the observations.

We quantify the sampling uncertainty of the KGE score for the selected basins based on the statistical benchmark following the methodology of Clark et al. (2021). This applies the non-overlapping bootstrap method (Efron and Tibshirani, 1986) to calculate tolerance intervals and jacknife-after-bootstrap method (Efron, 1992) for the standard error calculation of those tolerance intervals.

This method applies bootstrap and Jackknife methods to estimate the standard errors and tolerance interval of KGE uncertainty. The tolerance interval is defined as the difference between the 5th and the 95th percentiles. We ran the gumboot analyses package (Clark et al., 2021) with a sample size of 500 during the evaluation period to calculate the results.

### 2.4.2 Comparison of streamflow estimates

To provide more context to the results in terms of general model performance, we compared the streamflow estimates from wflow_sbm to those of the study by Knoben et al. (2020). The study by Knoben et al. (2020) ran 36 conceptual models using the Modular Assessment of Rainfall-Runoff Models Toolbox V1.0 (MARRMoT) (Knoben et al., 2019a, b) on the CAMELS data set. First, we calculated the mean of the 36 models for each basin. Next, we ensured a match between the basins under investigation by both studies. Due to differences in time period, forcing, and numerical solvers, the results cannot be directly compared to those of this study. It does however provide context to the results.

The inter-model (instance) comparison of the streamflow estimates in this study is assessed using a cumulative distribution function (CDF). We applied the Kolmogarov-Smirnoff test (Kolmogorov, 1933; Smirnov, N.V., 1933) to test if the differences between the KGE score distributions of the model instances are statistically relevant. This allows the acceptance or rejection of the hypothesis stating that the differences in streamflow estimates at various spatial resolutions will be small.

### 2.5 eWaterCycle platform

This research was conducted within the eWaterCycle platform (Hut et al., 2021). eWaterCycle follows by design the FAIR principles of data science (Wilkinson et al., 2018) and allows high level communication with models regardless of programming language through the Basic Modelling Interface (Hutton et al., 2020). This study showcases how eWaterCycle handles the setup of extensive modelling studies. A Jupyter Notebook with the model experiments of this study is provided in the GitHub

repository. As notebooks are not ideal for long-running experiments on high performance computing (HPC) machines, we exported the notebooks to regular python code which we ran directly on a HPC. The calibration and evaluation procedures totalled 41.025 model runs on the Dutch national super-computer Cartesius hosted by Surf.

## 3  Results

The results in this section are based on the modified KGE 2012 objective function applied to the streamflow estimates at the basin outlet. The Nash-Sutcliffe Efficiency (NSE) (Nash and Sutcliffe, 1970) results are available in the repository (DOI:10.5281/zenodo.5724576).

### 3.1  Calibration period results

#### 3.1.1  The effect of calibration on streamflow estimates

To illustrate how model calibration effects the streamflow estimates of each model instance, we first show the calibration curves of a single basin (ID:14301000). To avoid presentation bias we selected a basin with moderate performance and only show the last year of calibration. Figures 3abc show the calibration curves (yellow to red) for each of the instances that were generated by tuning the horizontal conductivity (i.e. KsatHorFrac) parameter. The parameter values range from 1 to 1000 and are a single value for each basin. Not all calibration interval values are shown in the figure for visualization purposes. The results depict the effect that increasing the KsatHorFrac values has on the hydrograph: base flow increases whilst peak flow reduces. In addition, large KsatHorFrac values reduce the flashiness of the streamflow estimates as is visible in Figures 3b and 3c in the second week of November 2005. Of note is that the selection of best calibration parameter values is strongly dependent on the chosen objective function as for example the NSE score would be more favourable towards flashiness and less towards base flow. As shown in Figure 3d the streamflow estimates of the model instance are similar (KGE score 0.58-0.66) while the parameter values deviate (KsatHorFrac 125-1000). No strong apparent trends for KsatHorFrac values in relation to model resolution or geographic location were found after calibration.

Figure 4 shows the CDFs of the KGE score distribution based on the best performing calibration model run of the 3 model instances. Starting with the modified KGE score distributions in Figure 4a we find that the model instance distributions are very similar for the whole domain, especially the 1km and 200m instances. The 3km instance has lower scores for 60 % of the distribution between 0.2 and 0.8 CDF. Approximately 18 % of the distributions is lower than a KGE score of -0.41 which corresponds to taking the mean flow. All three model instances show a similar Pearson r correlation CDF (Figure 4b) and gamma variability ratio (Figure 4c). The largest differences are visible in the beta bias component (Figure 4d that shows similar bias for the 1km and 200m with 60 % of the distribution lower than a value of 1 and 40 % higher. With the highest agreement in the lower and upper 5% of the distribution. The 3km instance agrees only in the upper 10% of the distribution with the 1km and 200m instances with 70 % of the 3km distribution lower than a value of 1 and 30 % higher. The bias term

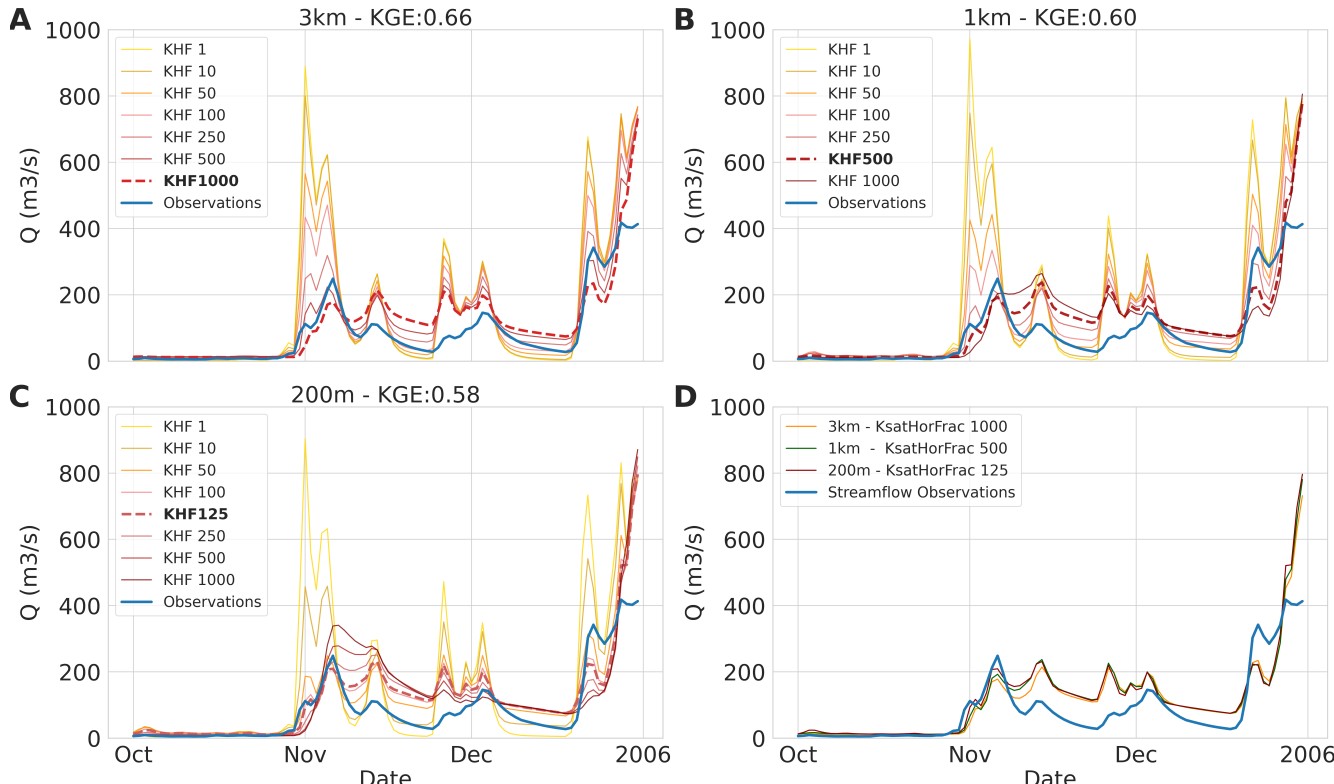

**Figure 3.** Calibration period results. The calibration interval of the KsatHorFrac parameter (KHF) for the 3 model instances at the basin outlet (ID:14301000) of the final calibration period year. (a) 3km model instance. (b) 1km model instance. (c) 200m model instance. Values range from 1 to 1000 (yellow to red), only a sub-selection is shown. Best performing calibration values are indicated with a dotted red line and streamflow observations in blue. (d) Best streamflow estimates of the 3km (orange), 1km (green), and 200m (red) model instances.

of the KGE score has the largest weight in determining the shape of the KGE score CDF as is shown by the 20 % of the
distributions with larger than 2.0 bias values.

## 3.2 Evaluation period results

### 3.2.1 Benchmark selection

The statistical benchmark is applied to determine which basins contain the streamflow estimates of the model instances that are deemed adequate for further analyses. The statistical benchmark is based on the best performing type of climatology of calendar day, either mean or median during the model evaluation period. Figure 5a shows spatially which type of calendar day benchmark is best performing per basin. Of the 567 simulated basins the results of 454 basins exceed the benchmark for each model instance. This is the case for all basins in the Midwest of the United States. Poor performance in comparison to

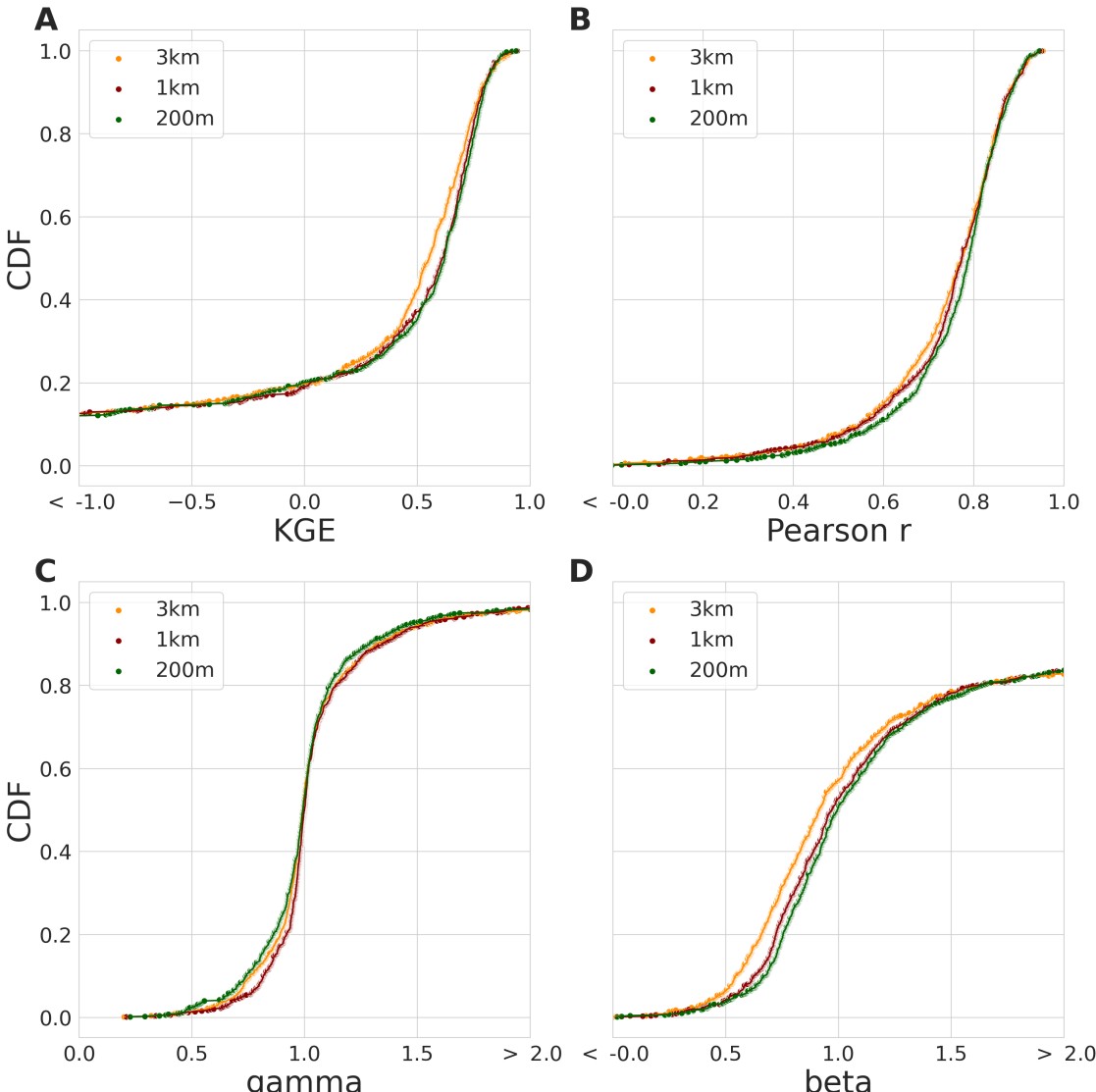

**Figure 4.** Calibration period results. The modified KGE score CDFs of the best performing model runs during the calibration period and its individual components of 3 model instances. With in orange the 3km, in red the 1km, and in green the 200m model instances. (a) The CDF of the modified KGE score. (b) The CDF of the Pearson r correlation component. (c) The CDF of the gamma (variability ratio) component. (d) The CDF of the beta (bias ratio) component.

the benchmark is mainly present in the Southwest. Based on the KGE scores, 83 % of the benchmark is favourable towards using the climatological mean and 27 % towards the median. An overview is provided in Figure 5c and the distribution of the benchmark KGE scores is shown in Figure 5d. The distribution ranges from -0.62 to 0.71 KGE score and is skewed towards values lower than 0.0 with a mean of 0.02 and median of -0.02.

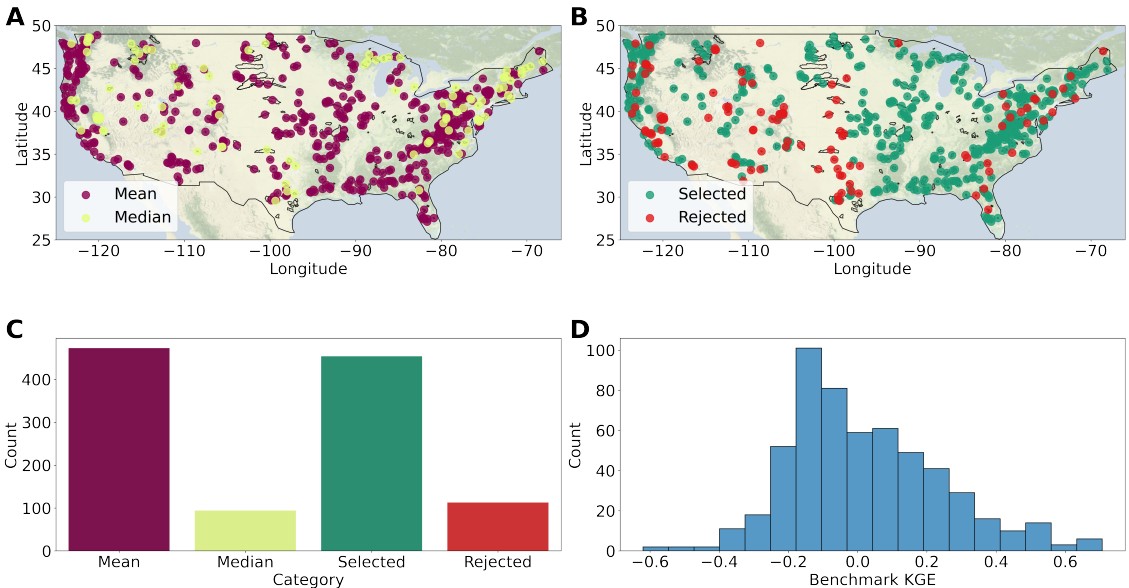

**Figure 5.** Evaluation period results. (a) The best performing type of 10 year calendar day climatology, either mean (purple) or median (yellow). (b) The spatial distribution of accepted (green) and rejected (red) basins based on the benchmark. (c) Overview of the amount of basins that are accepted or rejected and the best performing benchmark type. (d) The KGE 2012 score distribution of the best performing benchmark type. Basemaps are made with Natural Earth.

The statistical benchmarks (mean and median during the evaluation period) are plotted as CDFs based on the KGE score and its 3 individual components in Figure 6. For most of the KGE distributions in Figure 6a the mean benchmark outperforms the median with the upper 10 % being the exception. The Pearson correlation coefficient (Figure 6b) is slightly higher for the median at 70 % of the distribution. In the case of the gamma variability ratio (Figure 6c) the bottom 38 % is lower for the median than the mean and the upper 55 % slightly higher for the median than the mean. The determining component of the KGE score is the beta component shown in Figure 6d. As expected the mean benchmark is for the most part close to a value of 1, meaning that it is close to the mean of the observations. Of interest are the points of the median that are not close to 1, the lower 10% and the upper 1% of the distribution. These basins have flow regimes that greatly differ per year compared to the climatology. Considering mean statistical benchmark of basins with a beta bias score lower than $< 0.75$ and higher than 1.75. We find that the model instances outperform the statistical benchmark in 35 out of the 44 basins.

### 3.2.2 The effect of spatial resolution on streamflow estimates

The same 3 example basins as with the calibration period are used to illustrate the differences in streamflow estimates between model instances for the evaluation period. Only the last year of the evaluation period is shown in Figure 7.

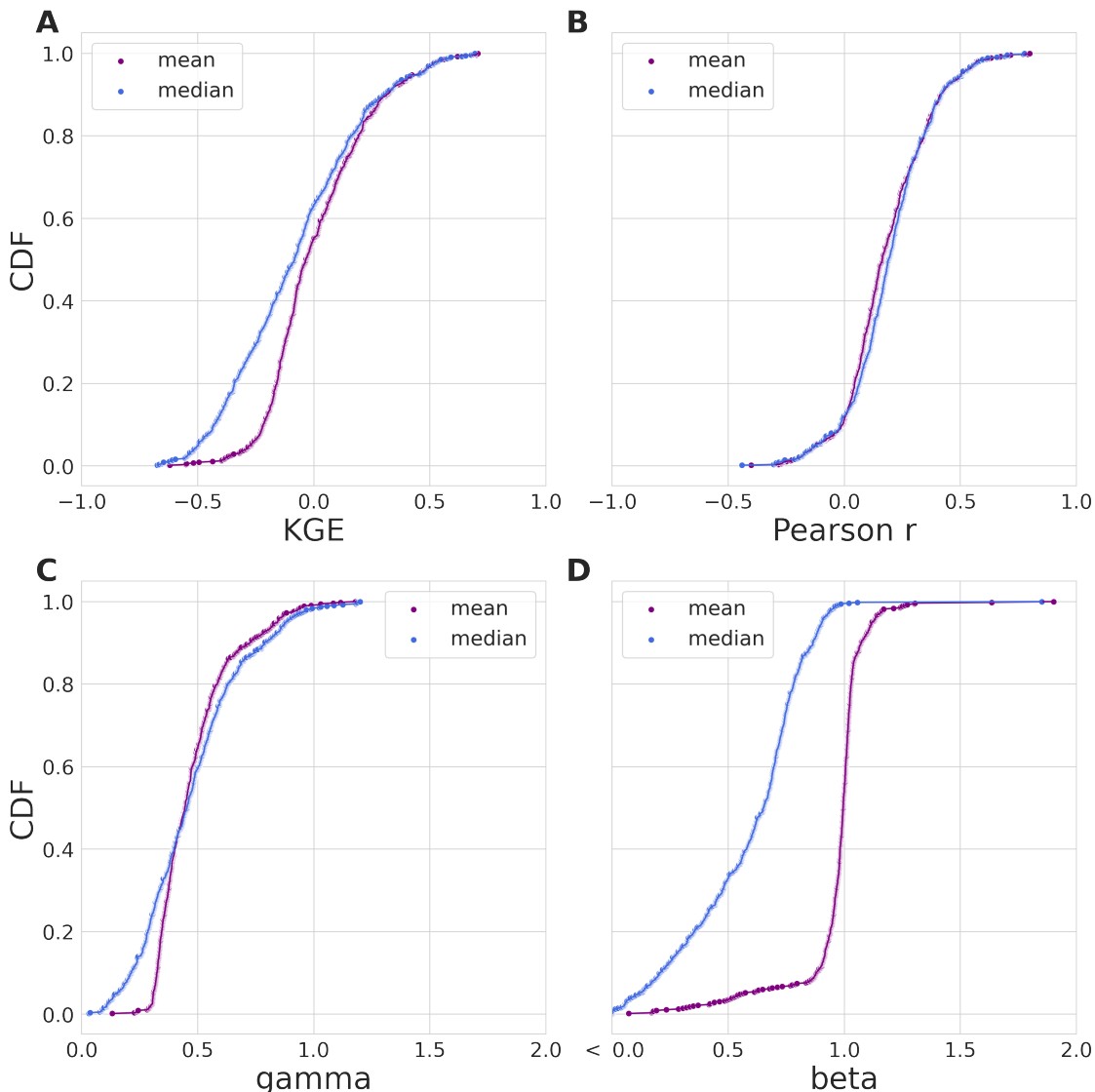

**Figure 6.** Evaluation period results. The CDFs of the modified KGE score and its 3 individual components of the statistical benchmark during the evaluation period. With in purple the mean and in blue the median statistical benchmark. (a) The CDF of the modified KGE score. (b) The CDF of the Pearson r correlation KGE component. (c) The CDF of the Gamma (variability ratio) KGE component. (d) The CDF of the Beta (bias ratio) KGE component.

In the case of poor performance, Figure 7a, it may occur that the model instances are overestimating streamflow during peak flow. The best performing model instance has the smallest peak flow estimates which in many cases is the coarsest spatial resolution instance (3km). Of the 454 basins, 3km instance has the lowest peakflow estimates in 279 occurrences of which the model is best performing 148 times. The example in Figure 7b shows that when the 1km model instance is best performing this

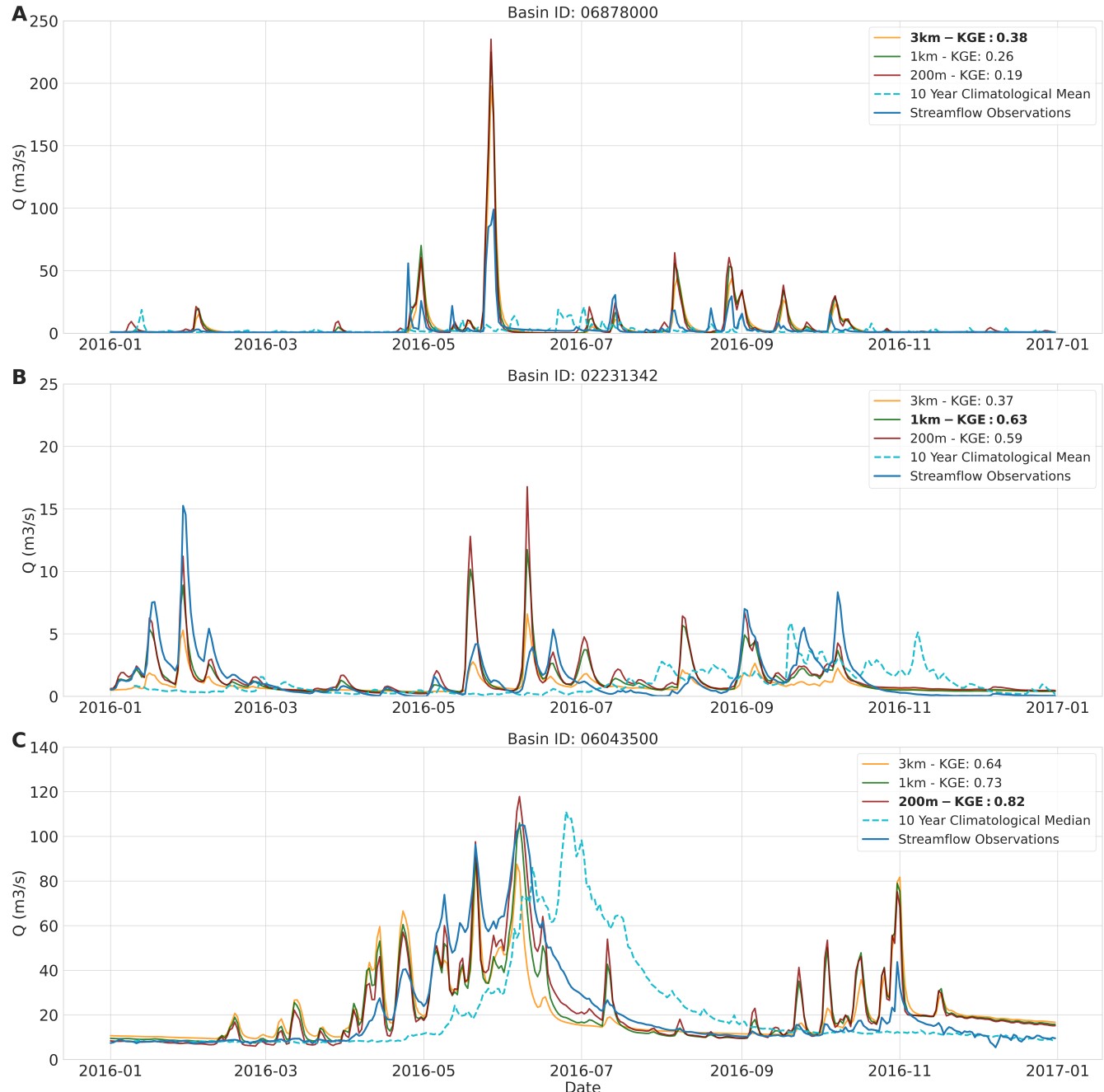

**Figure 7.** Evaluation period results. 3 example hydrographs showing the last year of the evaluation period. The 3km (orange), 1km (red), and 200m (green) model instance streamflow estimates at the basin outlet are shown. In blue the streamflow observations and in dotted cyan the 10 year calendar day climatology of the statistical benchmark. (a) Basin ID: 06878000. (b) Basin ID: 02231342. (c) Basin ID: 06043500.

often occurs in conjunction to relatively similar performance of the 200m model instance. In 78.7% of the cases where the 1km instance is best performing the difference in KGE score with the 200m instance is smaller than 0.1. The final example in Figure 7c illustrates when the finest spatial model resolution (200m) is best performing and the coarsest (3km) least performing. The 200m model instance best captures peak flow and the receding limbs of the hydrograph.

### 3.2.3 Streamflow estimates of model instances

The KGE score results for the evaluation period are shown in Figure 8a. The KGE score distribution of the mean of 36 hydrological models from Knoben et al. (2020) is included and is referred to as "MARRMoT mean results". Of note is that comparison between studies is not one-on-one due to differences in model run periods, forcing, and numerical solvers. We can, however, obtain information about general model performance between both studies.

The mean KGE score distribution of the MARRMoT models (Figure 8, blue) of Knoben et al. (2020) is close to the mean of the distributions of the 3 model instances. Differences between study results are mainly present in the tails of the distributions. Below 0.17 of the CDF (worst 17 % of the results) the MARRMoT mean results KGE score distribution is higher than the 1km model instance. The MARRMoT mean results for the lower 5 % of the CDF performs better than the 3 model instance distributions. Here, the range of KGE scores is smaller for the MARRMoT mean (-1.55 to 0.09) than for the 3 model instances (-13.56 to 0.00). Above 0.17 of the CDF (83 % of the results) the distributions of the 3 model instances are higher in KGE scores than those of the MARRMoT mean.

When we consider only the wflow_sbm instances, approximately 64 % of the results of the model instances are higher than 0.50 KGE score and of those 18 % are higher than 0.75 KGE score. The distributions cross at multiple points, for example at the bottom 10 % of the distribution the 3km instance has the highest and the 1km the lowest KGE score. At 40 % of the distribution and lower the 200m instance is followed by the 1km and 3km instances in terms of highest KGE score. The Pearson r component of the KGE score CDF in Figure 8b and the gamma variability ratio component 8c show small differences between model instances. The beta bias component in Figure 8d shows the largest differences between model instances. Especially between the 3km and the 1km and 200m model instances. The bias component is the main factor for the differences in the overall KGE score.

Next, we apply the Kolmogorov-Smirnov (KS) statistic to test whether the CDF of the model instances statistically differ from each other, for a given p-value of 0.05. The KS-test results in Table 2 show that the difference between 3km and 200m model instances is statistical relevant at a p-value of 0.02. On a large-sample, this means that increasing the spatial model resolution from 3km to 1km or 1km to 200m does not lead to significant differences in streamflow performance. When changing resolution from 3km to 200m, the distribution of KGE scores is significantly different (p < 0.05) according to the KS-test.

### 3.2.4 Objective function uncertainty

In addition to the streamflow evaluation we conducted a sampling uncertainty assessment of the KGE objective function using bootstrap and jackknife-after-bootstrap methods. The results of this assessment for each of the model instances is shown in Figure 9.The results for the 3 model instances are very similar. The tolerance interval results denoted with the black lines show

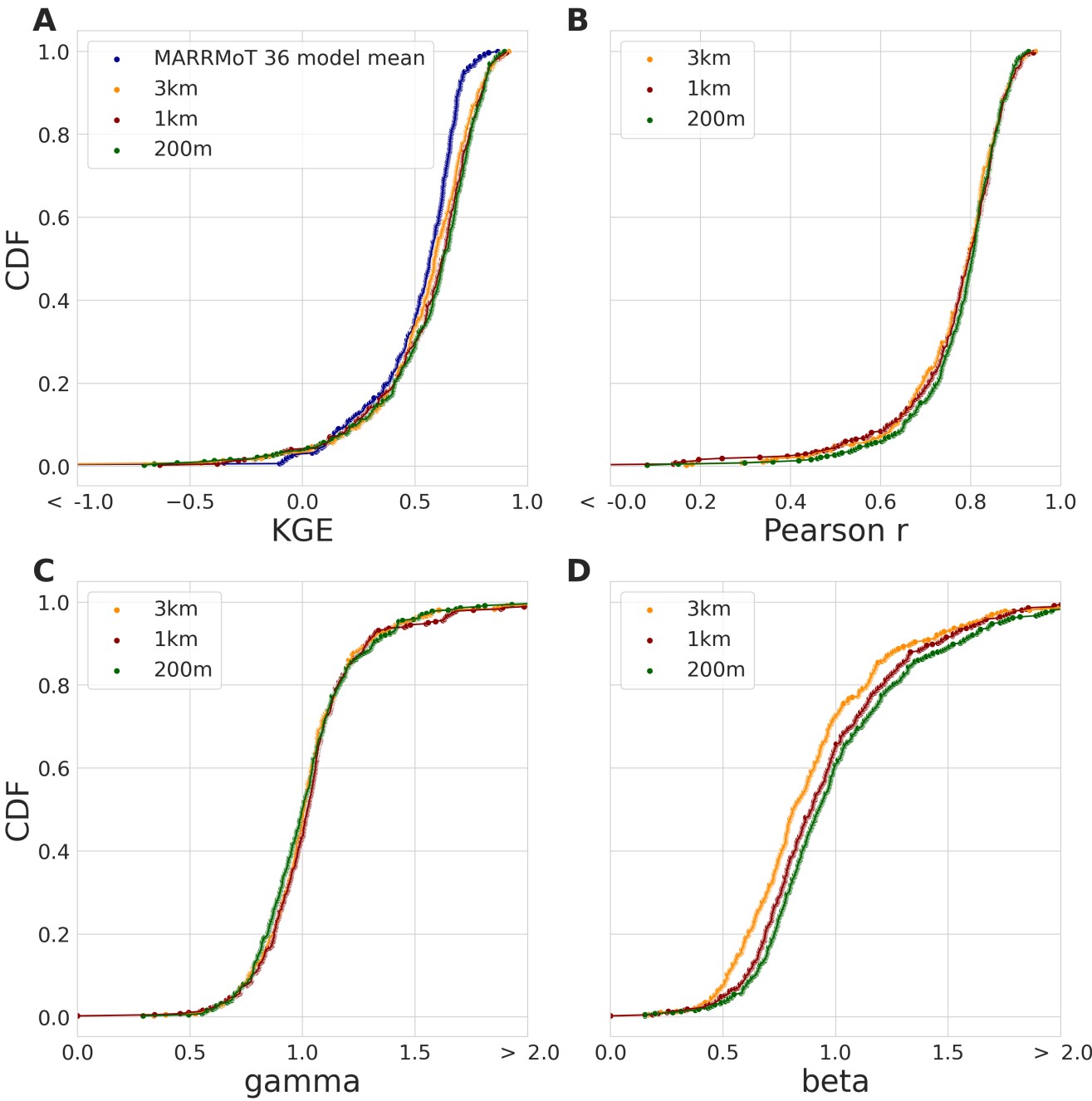

**Figure 8.** Evauation period results. The CDF based on the modified KGE scores and its 3 individual components for the 454 selected basins. (a) The modified KGE score with in blue the MARRMoT 36 mean results, in yellow the 3km, in red the 1km, and in green the 200m model instances. (b) The Pearson r correlation component. (c) The gamma variability component. (d) The beta bias component.

**Table 2.** The Kolmogorov-Smirnov statistic results and the corresponding p-value. The results are based on the difference between the KGE distributions of the 3 model instances, 3km, 1km, and 200m.

| CDFs | Kolmogorov-Smirnov Statistic | p-value |
|---|---|---|
| 3 km - 1 km | 0.08 | 0.14 |
| 1 km - 200 m | 0.05 | 0.82 |
| 3 km - 200 m | 0.11 | 0.02 |

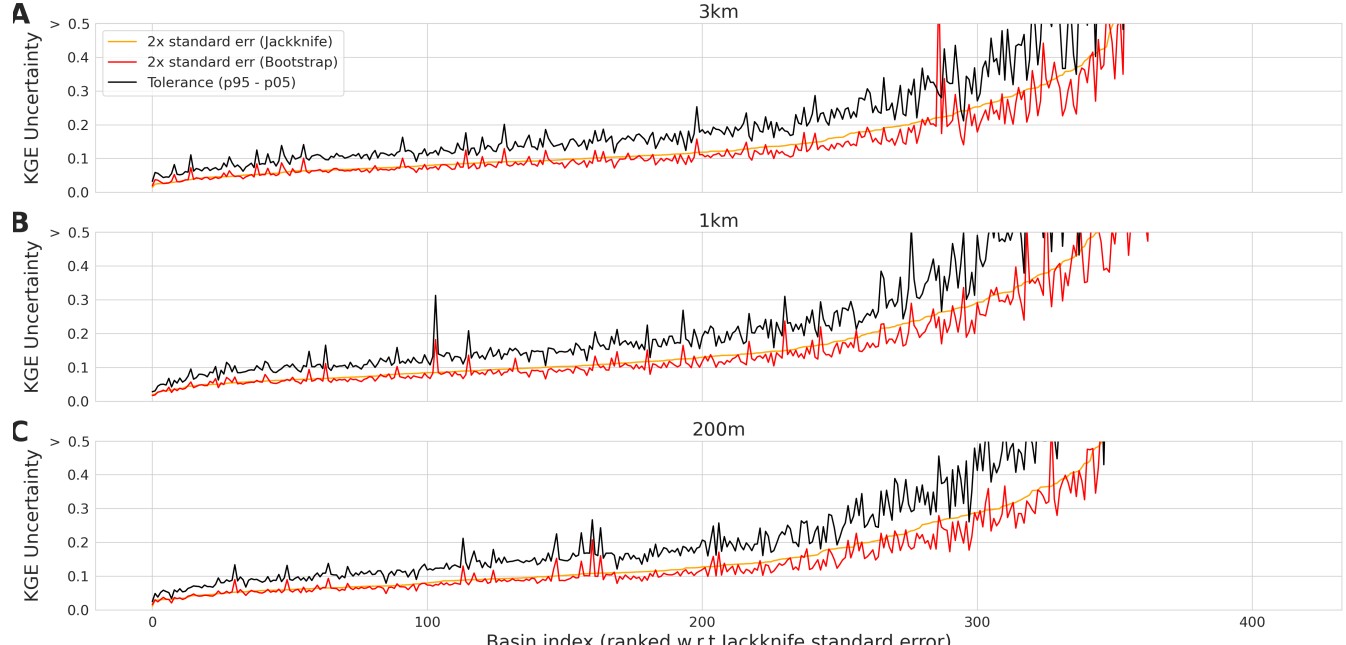

**Figure 9.** Evaluation period results. The bootstrap and jackknife-after-bootstrap results of the sampling uncertainty of the KGE score. In orange the 2x standard error of the jackknife method, in red the 2x standard error of the bootstrap method, and in black the tolerance interval obtained by subtracting the 5th percentile from the 95th percentile. The horizontal axis shows the basin index ranked w.r.t. the jackknife standard error and the vertical axis shows the modified KGE score sampling uncertainty. a. The 3km model instance results. b. The 1km model instance results. c. The 200m model instance results.

that approximately a 100 basins have a KGE sampling uncertainty of 0.1 or lower and approximately half of the basins 0.2 or lower. Half of the basins show high KGE uncertainty of more than 0.2 with approximately 80 basins surpassing 0.5 KGE for all model instances.

We project the sample uncertainty results on the CDF of the evaluation period 8 by calculating the mean of the tolerance interval, jackknife standard error, and bootstrap standard error for each of the 3 model instances per quarter of the total CDF. The results in Table 3 show slightly higher results for quarters of the cdf for the tolerance interval relative to the jackknife and bootstrap standard errors. The lower tail of the CDF contains the highest average values for the three sample uncertainty

**Table 3.** Sample uncertainty analyses results per quarter of the total percentage of the modified KGE cumulative distribution function of the evaluation period. The mean of the 3 model instance results is calculated based on the tolerance interval, jackknife standard error, and the bootstrap standard error, for each quarter of the total percentage.

| CDF | Mean Tolerance (p95-p05) | Mean 2x std err Jackknife | Mean 2x std err Bootstrap |
|---|---|---|---|
| 0.00 - 0.25 | 6.35 | 5.11 | 4.00 |
| 0.26 - 0.50 | 0.29 | 0.18 | 0.21 |
| 0.51 - 0.75 | 0.19 | 0.12 | 0.13 |
| 0.76 - 1.00 | 0.14 | 0.09 | 0.10 |

statistics and the upper tail the lowest average values.Indicating that sample uncertainty is high at low KGE values and vice versa.

### 3.2.5 Spatial distribution of evaluation period results

The CDF does not provide information at a basin level. To gain insight into the spatial distribution of the KGE scores of the model instances, Figure 10 shows the KGEs of the streamflow estimates plotted on a map of the CONUS domain. The minimum KGE scores of 0.50 to 0.89, shown in Figure 10a, are found in the Pacific Northwest, Atlantic South, Appalachia, and Northeast of the CONUS. KGE scores lower than -0.41 are found throughout the CONUS. The highest KGE scores in Figure 10b are located in the Northwest, Rocky Mountains, and Appalachia. These regions are characterised as steep sloping headwater basins. Figure 10c shows that there are large local streamflow discrepancies of more than 1.00 KGE score. These are mainly found in the Pacific Southwest, the South, and the Midwest. These regions span a wide range of hydro-climatic diverse basins. The average KGE score difference is 0.22. Figure 10d shows the best performing model instance for each of the 454 selected basins. Although regions show clusters in best performing model instance there is no overall geographical trend in results. Best performances for the 1km model instance are generally close to basins where the 200m model instance is best performing. The 3km model instance shows some clusters in the South and Pacific Northwest. The Rocky Mountains contains the best performing 200m model instances and the Appalachia a mixture between 1km and 200m model instances.

### 3.3 The effect of spatial scale on terrain characteristics

We illustrate the effect of spatial scaling on the parameter set of the 3 model instances by showing the difference in topography and drainage density for 3 basin. To avoid presentation bias, the basins were sampled based on poor streamflow performance (ID:06878000), moderate performance (ID:02231342), and good performance (ID:06043500). Figure 11a shows the probability density function (PDF) of the height distributions of the model instances for each of the 3 basins, Figure 11b shows the slope distribution, and Figure 11c the profile curvature distributions.

The height distribution of the DEM in Figure 11a shows, most clearly for basin ID 02231342, how the representation of the highest altitudes is underestimated by the 3km model instance (orange) compared to the 200m instance (green) and to a lesser

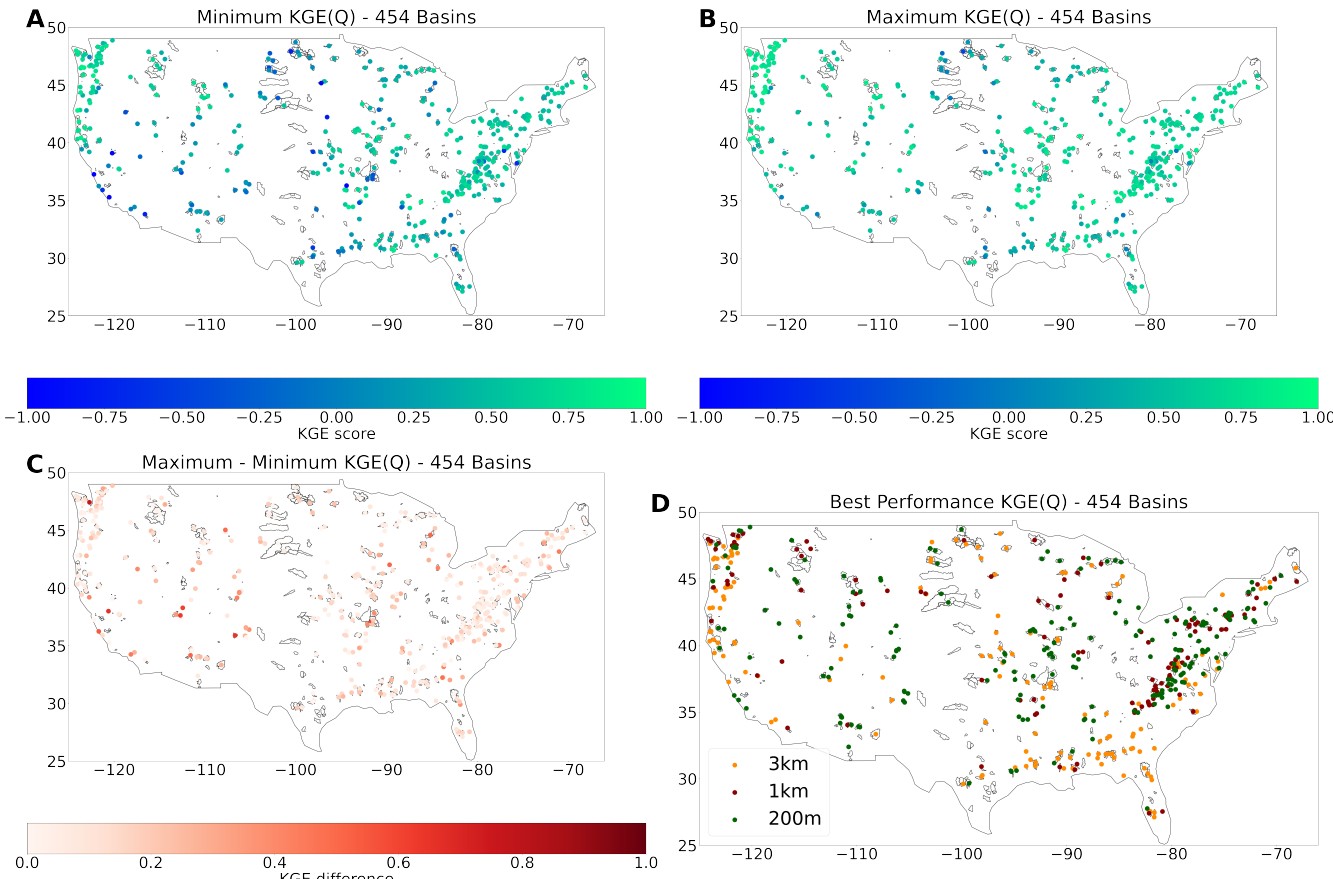

**Figure 10.** Evaluation period results. (a) Minimum KGE score of the model instances. (b) Maximum KGE score of the model instances. (c) The difference between minimum and maximum KGE scores. (d) Best performing model instance based on KGE score for the evaluation period with 3km in orange, 1km in red, and 200m in green.

extent the 1km instance (red). Essentially, at coarser spatial resolution the terrain is flattened at high altitudes. An opposite effect is shown in this basin for the lower altitudes where the finer resolution instances better capture gentle slopes that are flattened at coarse resolution. This effect is also detectable in the slope and profile curvature PDFs shown in Figures 11bc. As

can be expected from the height distributions, the slope of the 200m instance has more gentle and steep sloping topography than the 1km and 3km instances. This is shown by the narrower slope distribution for the coarse spatial resolution that broadens with finer resolution. The differences in the mean slope of the basins between model instances is marginal, e.g. 0.00019 m*m-1 for basin ID 02231342. The profile curvature in Figure 11c indicates whether a slope is linear (values close to 0), concave (values smaller than 0), and convex (values larger than 0). The 3km and 1km instances show similar slope geometries. With the

3km instance having slightly more linear slopes. At the finest resolution (200m) the slopes geometry shifts from linear slopes to either convex or concave curvature profiles.

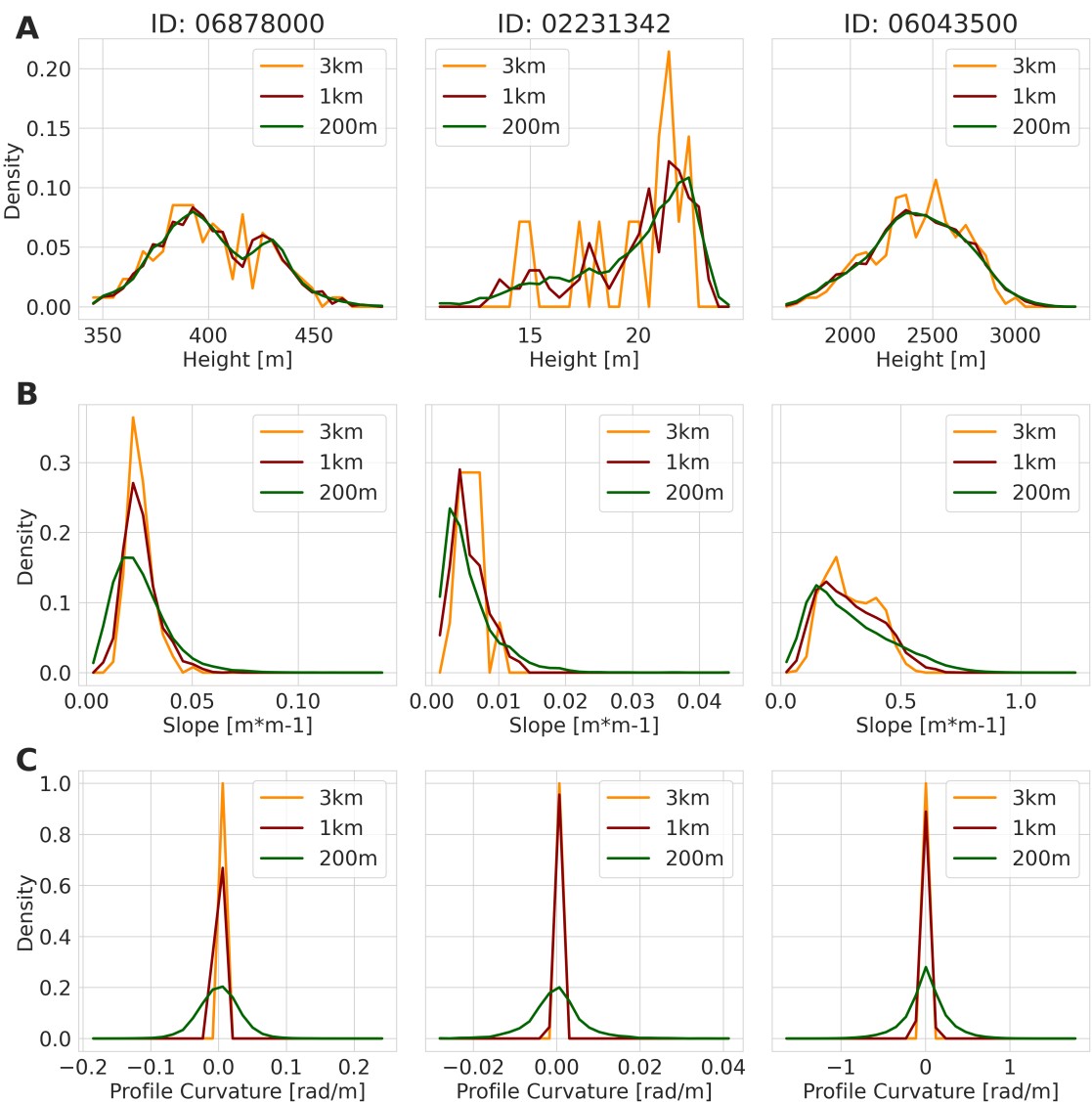

**Figure 11.** 3 example basins that represent poor streamflow performance (ID:06878000), moderate performance (ID:02231342), and good performance (ID:06043500). (a) PDF of height distribution for the 3km (orange), 1km (red) and 3km (green) model instances. (b) PDF of the slope distributions of the model instances per basin. (c) PDF of profile curvature distributions of the model instances per basin. Values that equal 0 indicate linear slope geometry, smaller than 0 concave slope geometry, and larger than 0 convex slope geometry.

In addition to topography we calculated the drainage density for each of the model instances defined as total river length divided by basin area. The results in Table 4 show small differences between the model instances for each of the 3 basins.

**Table 4.** The drainage density defined as stream length divided basin area for the 3 example basins.

| Basin ID | 3 km Drainage Density [m*m-2] | 1 km Drainage Density [m*m-2] | 200 m Drainage Density [m*m-2] |
|---|---|---|---|
| 06878000 | 0.0133 | 0.0141 | 0.0143 |
| 02231342 | 0.0056 | 0.0059 | 0.0063 |
| 06043500 | 0.0112 | 0.0123 | 0.0126 |

## 4 Discussion

### 4.1 Benchmarks

We applied an initial statistical benchmark based on streamflow observations for basin selection to identify basins which streamflow estimates are deemed adequate for further analyses. This does not imply that excluded basins are less relevant. Instead it implies that an in-depth model assessment is required to understand why the model is not able to simulate adequate streamflow estimates in these basins. The CDFs of the KGE score and its components based on the statistical benchmarks in Figure 6 show that benchmark is relatively easy to beat given the KGE score distributions. We find, as one might expect, that the beta bias component has values close to 1 for the mean benchmark while this is not the case for the median benchmark. Bias values not close to 1 for the mean benchmark indicate that the flow regime changes from year to year. The hydrological model simulation were in most cases able to capture this change better than the benchmark. This shows that the model is able to capture yearly variability and is not overfitting due to extensive calibration. In addition to the benchmark, we added a layer of context by including results from the study by (Knoben et al., 2020). This is an imperfect comparison due to differences in model inputs, numerical solvers, and simulation period. However, the results do provide information on general model performance. The results show that the wflow_sbm streamflow estimates are inline with estimates of the mean of the 36 MARRMoT models. The spread of results is smaller for the 36 MARRMoT models which is due to averaging and likely the more extensive calibration routine of the conceptual models. It also implies that when only streamflow at the basin outlet is under consideration users should carefully consider various model structures before model selection due to the small differences in results at the system-scale.

Other studies have conducted large domain modelling efforts with the CONUS as case study area (e.g. Mizukami et al., 2017; Rakovec et al., 2019). However these are hard to compare with the results from this study as they did not use the same basins. To improve future comparative work, we advocate for the creation of model output storage guidelines that use the CAMELS data set as case study area. These guidelines should encompass the differences between hydrological models, such as distributed and non-distributed modelling grids. A first step is the inclusion of distributed data sources in the CAMELS data set, e.g. meteorological data. This can be extended by including model evaluation products such as snow cover and soil moisture for further benchmarking. We further propose the use of a model experiment environment, such as eWaterCycle (Hut et al., 2021) to generate model results. This allows for similar pre-processing of inputs, standardization of outputs, and reproducible

modelling studies. An example of how to apply these steps using the eWaterCycle platform is provided in the Jupyter Notebooks that supplement this publication (DOI:10.5281/zenodo.5724512). The ease of setting up a model experiment and storing output is an incentive for users to store model results while conducting extensive modelling studies even when results are deemed not suitable for publication that might still benefit the community.

## 4.2 Streamflow estimates and uncertainty

At the start of the study we hypothesized that differences between model instances would be small due to quasi-scale invariant parameter sets and constant hydrological process descriptions within the model. The results of the calibration period in Figure 4 and the evaluation period in Figure 8 show that this is the case for the KGE score distributions based on streamflow estimates at the basin outlet. Although the differences are small, the crossing of the distribution lines is a strong indication that there is disagreement on KGE scores between model instances and that there is no single instance outperforming the rest consistently.

The largest difference between distribution for both periods is found in the beta bias component of the KGE score. The benchmark selection excluded mainly the basins that contained large bias values during the calibration period. For the bias component, the 3km instance deviates from the 1km and 200m instances. This shows that the total volume of streamflow does differ for the 3km instance. This difference was confirmed by testing the statistical significance of the differences between distributions of the model instances by applying the KS-statistic on the evaluation period results. Here a statistical difference

between distributions is only found for model instance combination 3km and 200m Table 2.

This study applies a single objective function, the modified KGE (Kling et al., 2012), to determine simulated streamflow adequacy for the model calibration and evaluation time periods. To provide the reader with more context we have included the KGE 2009 (Gupta et al., 2009), modified KGE , KGE non-parametric (Pool et al., 2018), and NSE objective function results in the data repository and in Figure A2. The conclusions based hypothesis testing were not affected by the type of KGE objective

function. We selected the modified KGE objective function as it is less influenced by extreme combinations of simulated and observed streamflow and less influenced by structural error in the meteorological input. The single objective function for a whole period approach is limited and can be improved by first determining the objective function for individual years and then averaging for the whole period (Fowler et al., 2018). In addition, as stated in Clark et al. (2021) it is important to determine the sampling uncertainty of objective functions to avoid wrong conclusions at the system scale. Following their methodology we

investigated the sampling uncertainty by applying bootstrap and jackknife-after-bootstrap methods (Figure 9). Approximately half of the basins has a KGE uncertainty of 0.2 or lower based on the tolerance interval and half of the basins higher than 0.2 with values surpassing 0.5 KGE uncertainty. The lower 25 % of the cumulative distribution function (Figure 8 contains the highest sample uncertainty and the upper 25 % the lowest (Table 3). Given these results we argue that the differences found using the KS-statistic are easily within the sampling uncertainty margins and therefore not valid to base conclusions on.

More so, when taking into account the model uncertainty and the observation uncertainty. This demonstrates the drawbacks of conducting large-sample assessments and the sensitivity to sampling uncertainty of the results.

## 4.3 Relative model instance differences

We recognize that large-sample assessments obscure variations in simulations between instances due to the sample size. On a basin level we find that local variations due to spatial resolution are in effect throughout the domain. This is depicted by the differences between the KGE scores of the model instances (Figure 10c). On average these is a 0.22 KGE score difference with extremes of more than 0.5 KGE score difference at multiple basins. In this case we are interested in the relative differences between the instances to understand the effect of varying spatial resolution. According to Oreskes et al. (1994) the only form of validation that is actually possible. Therefore, we argue that the differences are large locally even though they might be within the sampling uncertainty range.

We find that the 1km instance is best performing in basins where the difference between minimum and maximum KGE score of the 3 instances is small (Figure 10c). We attribute this partly to the calibration routine finding a better optimal parameter value (KsatHorFrac) as results are often close to those of the 200m instance. For the best performing 3km or 200m model instances (Figure 10d) there are only small geographical trends of best model performance in the South and Appalachia. This information is valuable for future research that does a more in-depth assessment of the internal states and fluxes as we now know where the effects of varying grid cell size is small or large and which model is best performing.

We conducted a terrain analyses (Figure 11) to identify changes in terrain characteristics due to spatial resolution that might explain the differences between model instance streamflow estimates. Minor effects are depicted by the profile curvatures present at 200m resolution. Slopes are less linear at fine resolution than at coarse resolution. The effect on the hydrological response is, however, expected to be small as stated by Bogaart and Troch (2006). Similarly small changes are found for the differences in drainage density between model instances (Table 4). This confirms that the drainage network up-scaling method of Eilander et al. (2021) is (almost) consistent across spatial scale.

Larger differences between model instances are found for the height distribution of the DEM, which is flattened at course resolution compared to finer resolution. At high altitudes this introduces changes in snow dynamics due to the use of the temperature degree-day method by the hydrological model. The resulting effect on streamflow estimates depend on the relative contribution of snow melt. Although marginal at a basin level, the difference in slope between instances is expected to effect the partitioning of the lateral fluxes of the wflow_sbm model as the lateral connectivity between grid cells is slope driven. An increase in slope would lead to larger lateral fluxes and vice versa. Increasing spatial resolution, aggregating the DEM, results in a broader distribution of slopes that effects the volume and timing of streamflow estimates. The effect of terrain smoothing has been reported by Shrestha et al. (2015) and they found this to increase overland flow lateral flux. An in-depth assessment of the internal states and fluxes of the model instances is required to determine whether these components are the main cause for the differences in streamflow estimates.

We applied the same meteorological forcing products and pre-processing routine for each model instance. This ensured that the total volume of precipitation remained consistent across scales. A coarse grid cell contains a volume of precipitation that is equally redistributed over the equivalent amount in size of finer grid cells. In reality this redistribution of water might not be equal across the finer grid cells and therefore scaling behaviour is introduced due to the locality of precipitation. This

has an effect on the streamflow estimates as the locality of precipitation directly influences hydrological processes that are dominant at different locations (e.g. hill slope). In addition, due to the large difference between native data and model instance resolution it is likely that the effects of disaggregation of precipitation and temperature lapsing are main drivers for differences in streamflow estimates between model instances. However, Shuai et al. (2022) found by comparing the same forcing product with native implementations at various spatial resolutions that the effect on streamflow estimates was relatively small. This was not the case for distributed variables in the basin (e.g, snow water equivalent). It is of interest to investigate this effect on streamflow estimates and to determine the role of native spatial meteorological forcing resolution.

### 4.3.1 Computational cost

When we consider the increase in streamflow based model performance as opposed to computational cost, we find that this does not scale linearly with the amount of grid cells in the basin due to lateral connections in the hydrological model. The average non-parallel run time of the 3km instance is 157 seconds and that of the 200m instance 12120 seconds with an average number of grid cell difference of 28872 cells. These results point toward the importance of conducting an initial spatial model resolution assessment at the start of large-sample assessments as it avoids subpar or computationally expensive model runs. Note that this kind of information can stimulate scientific and/or computational developments, e.g. in the meantime the wflow code was rewritten in Julia (van Verseveld et al., 2021) roughly increasing performance by a factor of 3 while other improvement (threading, mpi) are being implemented. There are alternative approaches to the spatial discretization of basins that are computationally very efficient such as the vector-based configurations that have the added benefit to for example better capture topographic details and are less influenced by native forcing resolution ((Gharari et al., 2020)).

### 4.4 Outlook

The results from this study help model developers with model refinement by providing them with understanding where and under what circumstances difference due to spatial scaling occur. Based on the aggregated domain and basin level results we can conclude that increasing spatial resolution does not necessarily lead to better streamflow estimates at the basin outlet. The implications of the results for the user are that caution is advised when interpreting high resolution model output as this does not directly translate into better model performance. Moreover, the computational cost of increasing model resolution is not always warranted compared to increase in streamflow estimate based model performance.

We conducted this study as an initial assessment to follow-up with studying scaling effects in distributed hydrological models. As the sampling uncertainty results showed it is very hard to draw conclusions from a large-sample and future research should therefore consider a smaller subset of basins to explore scaling effects in more detail. In this study we did not investigate individual basins to avoid biased selection of case study areas. We suggest that future work investigates the basins that show large or small differences in model performance, lateral fluxes, and effect of terrain aggregation to be part of this subset. In addition, the evaluation should go beyond streamflow by using multiple evaluation data products (e.g, soil moisture, evaporation, gravitational anomaly), see Guse et al. (2021) for a recent overview. Conclusions might then be made on whether increasing spatial resolution leads to increased model fidelity or not. This should be tested using multiple forcing data sets to test the

robustness of the conclusion. With the inclusion of multiple timescales as discussed in Melsen et al. (2016) more information
can be obtained about the linearity of hydrological process descriptions in the model.

## 5  Conclusions

The aim of this study was to analyse the effects varying spatial resolution has on the streamflow estimates of the distributed
wflow_sbm hydrological model. Distributions of model instance KGE score results were tested for significant differences as
well as the sampling uncertainty. A spatial distribution assessment was conducted to derive spatial trends from the results. The
main findings of the study are the following:

– The difference in the distributions of streamflow estimates of the wflow_sbm model derived at multiple spatial grid
resolutions (3km, 1km, 200m) is only statistically significantly different between the 3 km and 200 m model instances
(P<0.05). This confirms at an aggregated level the hypothesis that differences between model instances are small due to
quasi-scale invariant parameter set and process descriptions that remain constant across scale in the hydrological model.
However, the sampling uncertainty of the KGE score proved to be large throughout the domain. Therefore we conclude
that the minute differences found between model instances are to small to base conclusions on.

– Results show large differences in maximum and minimum KGE scores with an average of 0.22 between model instances
throughout the CONUS. Providing valuable information for follow up research based on the locality of relative model
scaling effects.

– There is no single best performing model resolution across the domain. Finer spatial resolution does not always lead to
better streamflow estimates at the outlet.

– Changes in terrain characteristics due to varying spatial resolution influence the lateral flux partitioning of the wflow_sbm
model and might be an important cause for differences in streamflow estimates between model instances.

This study answered where locality in results are strong due to varying spatial resolution. Future research should conduct an
in-depth assessment of basins where differences in streamflow estimates and lateral fluxes are large due to spatial scale. This
will lead to a better understanding of why and under what conditions locality in spatial scaling related issues occur.

*Code and data availability.*  The software that supplements this study is available at: https://github.com/jeromaerts/eWaterCycle_example_notebooks.
The data that supplements this publications is avaialble at DOI:10.5281/zenodo.5724576

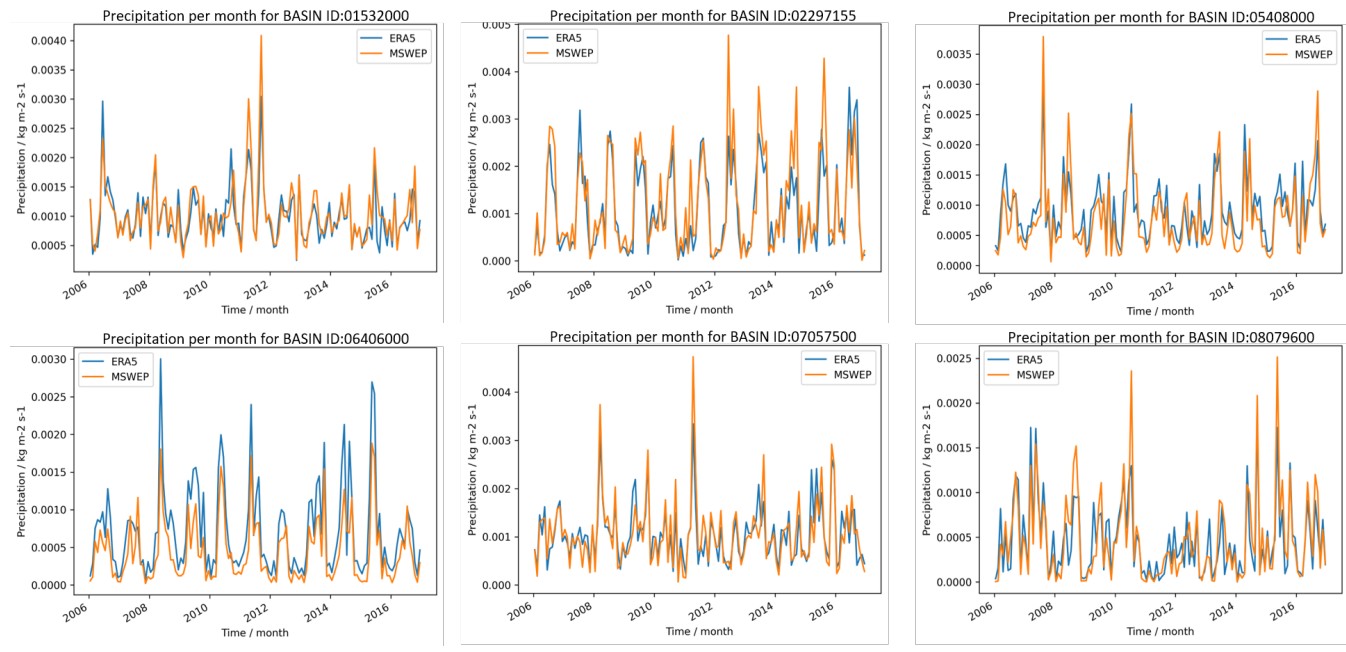

**Figure A1.** Evaluation period results. ERA5 and MSWEP forcing comparison for 6 basin in the CAMELS Dataset. Monthly precipitation values for the evaluation period are shown with in blue ERA5 and in orange MSWEP.

**Table A1.** Evaluation period results. Comparison of evaluation period objective function results of the 3km wflow_sbm instance based on the ERA5 and MSWEP forcing data sets.

| BASIN ID | Resolution | MSWEP - KGE 2012 | ERA5 - KGE 2012 | MSWEP - NSE | ERA5 - NSE |
|---|---|---|---|---|---|
| 01532000 | 3km | 0.20 | 0.19 | 0.67 | 0.65 |
| 02297155 | 3km | 0.23 | 0.04 | 0.34 | 0.06 |
| 05408000 | 3km | 0.35 | -0.03 | 0.60 | 0.39 |
| 06406000 | 3km | -1.69 | -6.19 | 0.21 | 0.15 |
| 07057500 | 3km | 0.68 | 0.53 | 0.56 | 0.35 |
| 08079600 | 3km | -5.65 | -8.13 | -.023 | -0.49 |

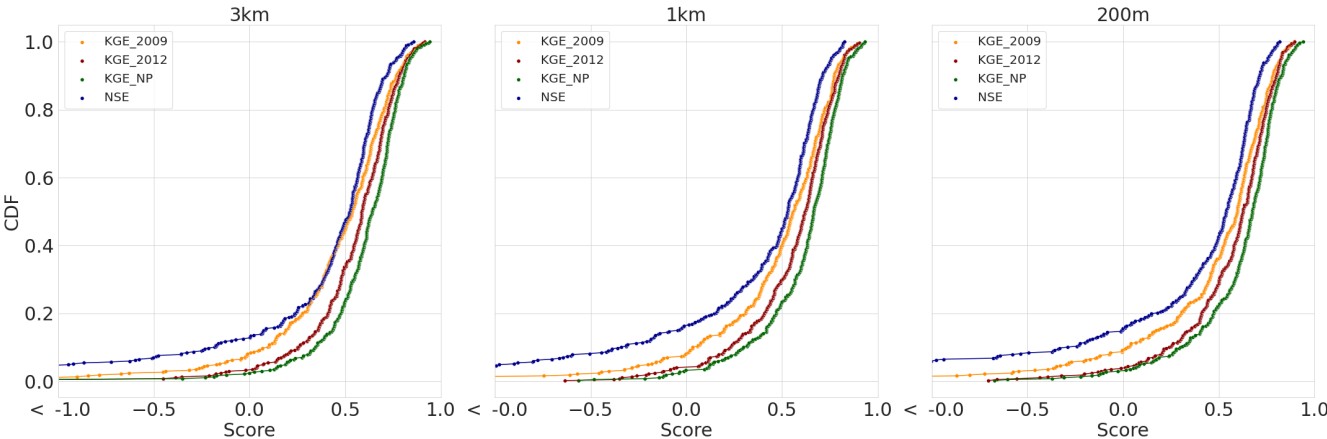

**Figure A2.** Evaluation period results. CDFs of multiple objective functions for the 3 model instances. With in orange KGE 2009, in red KGE 2012, in green KGE NP, and in blue NSE.

## Appendix A

### A1  ERA5 and MSWEP precipitation forcing comparison

### A2  CDFs of multiple objective functions

*Author contributions.* JPMA wrote the publication. JPMA, WJvV, AHW, PH did the conceptualization of the study. JPMA, ND, and PH developed the methodology. JPMA, WJvV, AHW, and PH conducted the analyses. RWH, NCvG, WJvV, AHW, and PH did an internal review. RWH, ND, and NCvG are PI of the eWaterCycle project.

*Competing interests.* The authors declare that no competing interests are present.

*Acknowledgements.* We would like to thank the anonymous reviewer and Shervan Gharari for their valuable feedback that helped improve this manuscript. This work has received funding from the Netherlands eScience Center under file number 027.017.F0. We like to thank the RSEs at NLeSC who have co-build the eWaterCycle platform and Surf for providing computing infrastructure.

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
