# Peer review of "Large-sample assessment of varying spatial resolution on the streamflow estimates of the wflow\_sbm hydrological model"

_Hydrology and Earth System Sciences, 2021_

## Author Comment (AC1)

**Response to Review #1**

Dear Referee #1,

We would like to thank you for reviewing our publication and for the constructive comments. As we believe that the comments will greatly increase the quality of the manuscript, we agree with all the suggestions and will adjust the publication accordingly. This will include the following points:

**Major Comments:**

*The streamflow performance of the different model instances is evaluated through the KGE score. Although the authors state in L150-L152 that they assessed the KGE score for both a calibration and an evaluation period, it seems that the results are mainly focused on the evaluation period: the CDFs of Figure 7 correspond to the evaluation period, and at least the map in Figure 8d also corresponds to the evaluation period according to the figure caption. It is not clear if Figures 8a, b and c also correspond to the evaluation period. The calibration results briefly appear in Figure 3 for an example basin, but I consider this insufficient. Therefore, my recommendation is to include the CDFs for the calibration period in Figure 7 (see also next two comments), and clearly distinguish between calibration and evaluation scores in the figure captions.*

We agree that adding information on the KGE values of the calibration and comparing it to the validation adds insight for the readers. The publication will include CDFs of the calibration period similarly to those of the evaluation period (Figure 7). These results will be placed in section 3.1 and include the three parts that compose the KGE score. In addition, a separate section in the discussion chapter discusses the effect of calibration given the insights gained by including the three components.

*Similarly to the NSE score, KGE can be decomposed into three parts: the coefficient of correlation, the ratio of the mean values and the ratio of the standard deviations (Gupta et al., 2007; Knoben et al., 2019). All these CDFs should be present in the manuscript, as they will help understand why the KGE values are as they are. Apart from the CDFs for KGE, Figure 7 should collect the CDFs for these three components (not necessarily for the MARRMoT ensemble, although it would be more than welcome). These new results should be discussed as well.*

Yes, we will adjust the figures to include the three components of the KGE score. These results will be included in the discussion. This will include the MARRMoT ensemble results.

*The two-fold statistical benchmark (one for the mean and one for the median) produces a poor performance (Figure 6d) that wflow_sbm can easily beat for most of the basins (Figure 6b). Although this is not a problem, I feel curious about why the KGE values are so low for the statistical benchmark. Then, the decomposition of the KGE score mentioned above should also be done for the statistical benchmark and should be incorporated into Figure 7 (a multi-panel figure where the plotted lines can be differentiated from each other may be the best way to show all this). This will help understand why the "mean statistical benchmark" outperforms the "median statistical benchmark" (Figure 6c). In particular, the ratio of the*

*mean values will provide an interesting insight: is the ratio of the mean values closer to one for the "mean statistical benchmark"?*

This is a very interesting point and will help explain the value of the statistical benchmark and its performance. We greatly appreciate the suggestion concerning the visualization of these results. This will be implemented accordingly for Figures 6,7, and the new calibration CDF figure. All figures will include the decomposed parts of the KGE score.

*The Discussion section is not structured and is written as a single block. It can be clearly divided into two parts: one part discussing the benchmark selection and one part discussing the spatial scaling effect. For sure, the new CDFs will strengthen the results and will enrich the discussion.*

We agree that the discussion section needs more structure and therefore we will subdivide the section. The subsection will follow the order of the results section and will have clear distinctions between results. This will help with the argumentation and the readability of the publication.

*I also miss in the discussion some recent and important references for the CONUS domain: for example, Mizukami et al. (2017) (already cited in the Introduction) and Rakovec et al. (2019) also carried out a large-domain calibration exercise and followed a benchmark approach to evaluate their results for the CONUS basins. Are the results of this study similar to their results?*

At an earlier research stage we have looked at these results and found "similar" behaviour between studies. We will include this in the discussion. This includes a comment on the differences between both studies.

**Minor Comments:**

*Title*

*The title is extremely long and sounds like a sentence extracted from the abstract or the conclusions. I would suggest a more concise title, something like "Large-sample assessment of spatial scaling effects on the streamflow estimations of a distributed hydrological model". The reader will find that "finer spatial resolution does not necessarily lead to better streamflow estimates" in the abstract. In any case, I will leave this open to the authors.*

Initially we decided on a "Nature Journal" style of title that describes the main conclusion of the study. Given this remark and that of Referee #2 we decided on restructuring such that the title is more in line with the HESS journal.

*Section 2.1.1 The CAMELS data set*

*The authors point out three reasons behind failed runs: errors during parameter derivation, errors during run time and missing streamflow observations. While the last one is clear, the other two are not properly described. What do the authors mean by "errors during parameter derivation"? Is this related to the parameter estimations from external sources prior to calibration? Or is it related to the calibration procedure? On the other hand, what do the authors mean by "errors during run time"? I suggest a more detailed description.*

This part needs more elaboration in the final publication. The errors stem from parameter estimation from external sources. These may occur during parameter estimation or come to the surface during runtime. Therefore they should be grouped together. This will be clearly described in the publication.

*Section 2.2.3 Model Runs & Calibration*

*The parameter KsatHorFrac is the only parameter subject to calibration, and the rest of the parameters are derived from external sources. Firstly, the parameter range for KsatHorFrac should be indicated here and not in L198 when the results are presented. Secondly, it is not clear if the selection of this parameter is based on prior studies, on calibration recommendations for wflow_sbm, or on a sensitivity analysis carried out by the authors. Some information is provided in L60-L62, but I find it confusing to read this in the introduction. I suggest mentioning this information in section 2.2.3 as I feel it belongs here.*

We will clarify the calibration routine by firstly stating the parameter range clearly. Secondly, by referencing previous sensitivity analyses (e.g., Imhoff et al. 2020). This information will be inserted in section 2.2.3.

*How is the model calibrated? Do the authors use a calibration algorithm? Is it based on a Montecarlo experiment? No details are given on the calibration procedure, only L153-L154 state that "the calibration procedure finds an optimal parameter value based on the KGE objective function of streamflow estimates at the basin outlet". The calibration procedure should be properly described.*

The model is calibrated "manually" by predefining a parameter range. The parameter corresponding to the highest KGE score at the basin outlet is then selected. The parameter range is selected based on the sensitivity of the parameter to the KGE score. We decided on manually calibrating the hydrological model as this greatly reduces the amount of compute time while still finding a close to optimal parameter value. This information will be included in the publication.

The following minor comments will be adjusted in the publication:

- *The last sentence in L187-L188 seems incomplete, or at least has no cohesion with the previous sentence.*

- *Instances of "Figure 7" throughout the paragraph seem to refer to Figure 6.*

- *"Figure 5" in L249 seems to refer to Figure 7.*

- *The colorbar in Figure 8c should indicate "KGE difference" or "âˆ † KGE". "KGE value" is not correct.*

- *Should "their" in L303 be "there"?*

Best Regards,

Jerom Aerts

---

## Author Comment (AC2)

**Response to Review #2**

Dear Shervan Gharari,

We would like to thank you for reviewing our publication and for the comments. We agree with many of the suggestions and will adjust the publication accordingly. In your review you also touch upon the larger discussion in the hydrological community on different model structures and on the software used for our study. We will separate the review of our work from the broader discussion as we believe that the focus on this platform should be on the assessment and improvement of the quality of the publication. We would like to point out that through accepting this work into the review process of HESS the editor has decided that it, in principle, fits within the scope of the journal and that the primary purpose of the review process is to test if the research is sound and if the manuscript can be improved. .

The response to the review is structured as follows: Part 1, response to comments that will improve the quality of the publication. Part 2, response to the ongoing discussion on the topic vector- and grid-based simulations. Part 3, response to the software used for the methodology of this study.

**Part 1: Response to comments regarding the publication**

- *References*

We agree that the reflection on the use of references should be (greatly) improved and that the references need to be extended to include the broader land-surface community past and present. This will include works on parameter identifiability such as Gupta & Sorooshian (1983), Grayson et al., (1992), and Oreskes et al., (1994). Works on parameter transferability over space such as Finnerty et al., (1997), Haddeland et al., (2002), Wagener and Wheater (2006), and Melsen (2016). A better description of the scaling/closure problem by including Wood et al., (1988), Bloschl and Sivapalan (1995), Beven (2006). This list will be extended in the revised publication.

- *Objective function relevance (KGE 0.22)*

This is a valid point that we will further address in the discussion section. In addition, we will extend the analysis by including methods presented in Clark et al. (2021) as per the Referee's suggestion. We believe that this will provide the reader with much needed reflection on the meaning of the differences in KGE score. The results will be incorporated in the discussion section of the publication.

- *Lakes and reservoirs*

We will report the presence of lakes and reservoirs in the model setup per subbasins. When relevant this will be included in the discussion section.

- *Basin selection*

The reason for selecting 454 CAMELS basins will be clarified as also per request of Referee #1. Reasons for excluding basins are errors that stem from parameter estimation from

external sources (mainly river network delineation related) and lacking streamflow observation records during the evaluation period of the model simulations.

- *Calibration methodology*

We will clarify the calibration routine by firstly stating the parameter range clearly. Secondly, by referencing previous sensitivity analyses (e.g., Imhoff et al., 2020; Wannasin et al., 2021). This information will be mentioned in section 2.2.3.

The model is calibrated "manually" by predefining a parameter range. The parameter corresponding to the highest KGE score at the basin outlet is then selected. The parameter range is selected based on the sensitivity of the parameter to the KGE score. We decided on manually calibrating the hydrological model as this greatly reduces the amount of compute time while still finding a close to optimal parameter value. This information will be included and clearly described in the publication.

- *Model selection*

In the publication we argue that the use of the wflow_sbm is of interest due to the premise of deriving a model that is semi-scale independent at various spatial resolutions from globally available data sources through the use of transfer functions. This is now more than relevant due to the trend in increasing the spatial resolution of grid-based models (e.g., Sutanudjaja et al., 2018). As these types of hydrological models are used for various applications we find it important to understand what this trend entails for users. In addition, we would like to note that the original Topog_SBM concept (Vertessy and Elsenbeer, 1999) was applied at a small scale.

- *Scaling in hydrology*

The focus in this study is on testing the effect that spatial model resolution has on the simulation of streamflow at the basin outlet on a large sample of basins and what this entails for users. This provides insights on the behavior of a hydrological model that holds the promise to derive comparable model instances at various spatial resolutions and therefore comparable streamflow simulations. Although we are not investigating the collective behavior of internal states and fluxes at various scales, we do investigate changes that occur in streamflow simulations due to variations in basin delineation, river network estimation, and topography (section 3.1). We consider these parts scaling from a hydrological model perspective.

- *Future research*

The Referee has made several valid points concerning the evaluation of the hydrological model. We hope these points will be addressed in future research (either by us or others in the community. For this publication the scope is on a single objective. Having said that, we do have some initial thoughts we want to share in this discussion:

- *Forcing and model resolutions*
  It is of great interest to investigate the effects that the native resolution of precipitation fields have on the simulations of the hydrological model. An initial assessment looked at applying stochastic downscaling using a climatology to (artificially) improve heterogeneity in the precipitation fields. Results on a small subset of the basins showed that this had small effects on the streamflow estimates at the outlet. However, in future

research we intend to further investigate the effect of native forcing resolution by including multiple forcing products (8km – 1km). In addition, we hope to investigate the effects of spatial resolution in relation to the numerics by applying uniform precipitation fields as a baseline, similar to Melsen et al. (2016).

- *States and fluxes, model fidelity.*
  We agree with the Referee that scaling in hydrology goes beyond adjustments of the model grid resolution. It is important to evaluate the collective behavior of the states and fluxes of the hydrological model at various scales to test whether parameterizations of processes are sufficient or not. This has been deemed out-of-scope and we invite the reviewers and the rest of the community to join us in working on this. This research should include a complete flux and state comparison and when possible states and fluxes are evaluated against (remote sensing) observations.

**Part 2: Response to comments regarding vector- vs grid-based simulations**

The discussion concerning grid-based simulation and vector-based simulation is an ongoing important discussion that can be held further during conferences and or commentaries. As the assessment of grid-based models is still relevant due to their application in operational and scientific settings, we consider analyzing such models as valuable to the hydrological community. Nonetheless, we would like to encourage a discussion on another platform on this topic with the Referee.

**Part 3: Response to the software used for the methodology of this study**

The work presented in this paper is one of the first to use the eWaterCycle platform for hydrological computational research. eWaterCycle is designed to let hydrologists be experts in hydrology without having to become a computer scientist in the process. As the reviewer rightly points out: There is a difference between users (hydrologists) and developers indeed. For any comments on the software of the eWaterCycle platform which is used in the workflow of this study we would like to refer you to the publication in GMD, https://doi.org/10.5194/gmd-2021-344. You are most welcome to write a reply to this study to start a discussion on the use of (Python) software packages and the philosophy behind eWaterCycle.

Best Regards,

Jerom Aerts

---

## Author Response (AR1)

Dear reviewers,

Here we provide a list of changes and a short point-by-point response to the reviews. Thank you again for the valuable feedback. We believe that this greatly benefitted the quality of the publication.

Kind Regards,

Jerom Aerts

List of changes:

1.  Title:
    Better describes content of the study and uses the term varying spatial resolution instead of scaling.
2.  Abstract:
    The abstract is updated based on the extra analyses we performed.
3.  Introduction:
    -   Restructured.
    -   Includes competing basin discretization approaches.
    -   Includes a larger body of cited literature that refers to the land surface community, parameter identifiability problem, parameter transferability, and the representative elementary watershed.
    -   Clearly stated that we are not using the MPR method.
    -   We now use "the effects of varying spatial resolution" instead of scaling as this is only a part of scaling.
4.  Methodology:
    -   Better description as to why basins are excluded from the analyses.
    -   Better description of the calibration methodology.
    -   Description of the sampling uncertainty of the KGE score method of Clark et al. 2021.
5.  Results:
    -   Restructured.
    -   Included calibration period CDF of the KGE score and the decomposed components.
    -   Added the evaluation period KGE score components to the CDF figure.

- Added objective function uncertainty section. Includes the analyses similar to Clark et al. 2021. Added a table summarizing these results based on the evaluation CDF.

6. Discussion:
   - Restructured. Added headers for readability.
   - Refer to other large domain studies.
   - Discuss the sampling uncertainty results and what this means for the statistical KS-test results.
   - Discuss the effect of using coarse meteorological forcing products.
   - Discuss how vector-based discretization has major benefits when it comes to computational cost and topographic discretization.

7. Outlook:
   - Expended on what needs to be added to this study in order to do a complete scaling assessment as opposed to only a spatial scaling assessment.

8. Conclusions:
   - State that the sampling uncertainty is large and therefore the conclusion based on the KS-test are inconclusive.
   - Added "at the outlet" to the conclusion regarding finer spatial resolution does not always leads to better streamflow estimates.

Point-by-point response to the reviews:

Review #1

- We included the calibration period CDFs of the KGE score and its individual components.
- We included the individual components of the KGE score with the evaluation period CDFs.
- We included the CDFs of the benchmark.
- We added structure to the discussion.
- We included important references for the CONUS domain (Mizukami et al. 2017; Rakovec et al. 2019) in the discussion.
- We resolved all the suggested minor changes.

Review #2

References

- We have extended the cited literature in the introduction by including the land surface community, parameter identifiability, parameter transferability, and the representative elementary watershed.

Lakes and reservoirs

- We reported the amount of lakes and reservoirs used by the model instances. Due to the small amount this did not alter the main conclusions of the study.

Basin selection

- Clearer description as to why basins were excluded from the analyses.

Calibration methodology

- Clearer description of the calibration methodology.

KGE sampling uncertainty

- We performed the suggested methodology of Clark et al. 2021 and discuss the original results in light of the sampling uncertainty.

Discretization:

- We discuss the benefits of vector-based discretization in the discussion.

Scaling:

- We refrain from using the term scaling and use varying spatial resolution instead. Added to the discussion and outlook what is required to analyse spatial scaling.

---

## Referee Report (RR1)

**Jerom P.M. Aerts et al. - HESS review**

All my suggestions have been satisfactorily addressed in the revised manuscript and I think it is ready for publication.

I particularly appreciate the decomposition of KGE into Pearson correlation, mean bias and variability bias, as they help understand why KGE scores are as they are for each calibration/evaluation experiment.

---

## Author Response (AR2)

Dear reviewers,

Here we provide a point-by-point response of the changes we made based on the reviews. Thank you for the valuable feedback. We believe that this greatly benefitted the quality of the publication.

Best Regards,

Jerom Aerts

**Point-by-Point response Review #1**

**Major Comments:**

*The streamflow performance of the different model instances is evaluated through the KGE score. Although the authors state in L150-L152 that they assessed the KGE score for both a calibration and an evaluation period, it seems that the results are mainly focused on the evaluation period: the CDFs of Figure 7 correspond to the evaluation period, and at least the map in Figure 8d also corresponds to the evaluation period according to the figure caption. It is not clear if Figures 8a, b and c also correspond to the evaluation period.*

We clarified the distinction between calibration period and evaluation period results by adding a first sentence to all relevant figure captions. In addition, we added the headers "3.1 calibration period results" and "3.2 evaluation period results" to add structure to the results section.

*The calibration results briefly appear in Figure 3 for an example basin, but I consider this insufficient. Therefore, my recommendation is to include the CDFs for the calibration period in Figure 7 (see also next two comments), and clearly distinguish between calibration and evaluation scores in the figure captions.*

We added Figure 4 that shows the KGE CDF of the calibration period results and the 3 KGE components. These results are described in lines 245 – 255 and included in the discussion.

*Similarly to the NSE score, KGE can be decomposed into three parts: the coefficient of correlation, the ratio of the mean values and the ratio of the standard deviations (Gupta et al., 2007; Knoben et al., 2019). All these CDFs should be present in the manuscript, as they will help understand why the KGE values are as they are. Apart from the CDFs for KGE, Figure 7 should collect the CDFs for these three components (not necessarily for the MARRMoT ensemble, although it would be more than welcome). These new results should be discussed as well.*

We have added the CDFs of the 3 KGE components to the newly created calibration period results Figure 4 (described in lines 245-255) and to the existing evaluation period results Figure 8 (described in lines 289-307). The new results are discussed in lines (395 – 401) of the discussion section.

*The two-fold statistical benchmark (one for the mean and one for the median) produces a poor performance (Figure 6d) that wflow_sbm can easily beat for most of the basins (Figure 6b). Although this is not a problem, I feel curious about why the KGE values are so low for the statistical benchmark. Then, the decomposition of the KGE score mentioned above should also be done for the statistical benchmark and should be incorporated into Figure 7 (a multi-panel figure where the plotted lines can be differentiated from each other may be the best way to show all this). This will help understand why the "mean statistical benchmark" outperforms the "median statistical benchmark" (Figure 6c). In particular, the ratio of the mean values will provide an interesting insight: is the ratio of the mean values closer to one for the "mean statistical benchmark"?*

We added the statistical benchmark CDFs of the KGE score and its individual components in Figure 6. The results are described in lines 267-276. Indeed this provided an interesting insight regarding the bias (mean values) component. These are closer to 1 for the mean statistical benchmark. This is discussed in lines 364-369.

*The Discussion section is not structured and is written as a single block. It can be clearly divided into two parts: one part discussing the benchmark selection and one part discussing the spatial scaling effect. For sure, the new CDFs will strengthen the results and will enrich the discussion.*

We have added much needed structure to the discussion section. The new structure is now as follows: 4.1 Benchmarks, 4.2 Streamflow estimates and uncertainty, 4.3 Relative model instance differences (spatial scaling), 4.4 Computational cost.

*I also miss in the discussion some recent and important references for the CONUS domain: for example, Mizukami et al. (2017) (already cited in the Introduction) and Rakovec et al. (2019) also carried out a large-domain calibration exercise and followed a benchmark approach to evaluate their results for the CONUS basins. Are the results of this study similar to their results?*

We have added the references for the CONUS domain in the discussion lines (377 – 379). Due to the many differences between studies we find it not possible to compare results. This is one of the reasons why we advocate for clear guidelines for modelling studies to facilitate future comparative work.

**Minor Comments:**

*Title*

*The title is extremely long and sounds like a sentence extracted from the abstract or the conclusions. I would suggest a more concise title, something like "Large-sample assessment of spatial scaling effects on the streamflow estimations of a distributed hydrological model". The reader will find that "finer spatial resolution does not necessarily lead to better streamflow estimates" in the abstract. In any case, I will leave this open to the authors.*

We changed the title to: "Large-sample assessment of varying spatial resolution on the streamflow estimates of the wflow_sbm hydrological model". Note that we

removed the term "spatial scaling" as reviewer #2 pointed out that a spatial scaling assessment goes beyond varying spatial resolution.

*Section 2.1.1 The CAMELS data set*

*The authors point out three reasons behind failed runs: errors during parameter derivation, errors during run time and missing streamflow observations. While the last one is clear, the other two are not properly described. What do the authors mean by "errors during parameter derivation"? Is this related to the parameter estimations from external sources prior to calibration? Or is it related to the calibration procedure? On the other hand, what do the authors mean by "errors during run time"? I suggest a more detailed description.*

We clarified the reasons behind failed runs (lines 100 – 103). Besides missing streamflow observations errors occurred due to parameter estimation errors that in some cases became clear during runtime. Therefore we now only state parameter estimation error as a reason. These occur during drainage network delineation, either when the basin outlet consisted of a single grid cell that results in a model coding error or when inconsistencies occurred in the local drainage direction layer.

*Section 2.2.3 Model Runs & Calibration*

*The parameter KsatHorFrac is the only parameter subject to calibration, and the rest of the parameters are derived from external sources. Firstly, the parameter range for KsatHorFrac should be indicated here and not in L198 when the results are presented. Secondly, it is not clear if the selection of this parameter is based on prior studies, on calibration recommendations for wflow_sbm, or on a sensitivity analysis carried out by the authors. Some information is provided in L60-L62, but I find it confusing to read this in the introduction. I suggest mentioning this information in section 2.2.3 as I feel it belongs here.*

We have added a clearer description as to why we calibrate the parameter KsatHorFrac in lines 164 – 171 in section 2.2.2 Model Runs & Calibration.

*How is the model calibrated? Do the authors use a calibration algorithm? Is it based on a Montecarlo experiment? No details are given on the calibration procedure, only L153-L154 state that "the calibration procedure finds an optimal parameter value based on the KGE objective function of streamflow estimates at the basin outlet". The calibration procedure should be properly described.*

The calibration routine is now more extensively described in lines 164 - 178. We manually calibrated the parameter based on an interval ranging between 1 – 1000 KsatHorFrac values. The best performing model run based on streamflow estimates at the outlet is evaluated using the modified KGE score and subsequently used for the evaluation period.

*The following minor comments will be adjusted in the publication:*

- *The last sentence in L187-L188 seems incomplete, or at least has no cohesion with the previous sentence*

- *Instances of "Figure 7" throughout the paragraph seem to refer to Figure 6.*

- *"Figure 5" in L249 seems to refer to Figure 7.*

- *The colorbar in Figure 8c should indicate "KGE difference" or "â⁁ † KGE". "KGE value" is not correct.- Should "their" in L303 be "there"?*

These minor comments have been resolved in the revised manuscript.

**Point-by-Point response Review #2**

***The review covered lots of ground and not always specific parts of the publication therefore we provide a more generic point-by-point response.***

**Comments:**

*References*

We agreed that the reflection on the use of references should be improved and that the references need to be extended to include the broader land-surface community past and present. Therefore we extended the cited literature in the introduction by including competing modelling philosophies, bodies of work from the land surface community, the parameter identifiability and transferability problems, and the representative elementary watershed concept. As mentioned in the review we adjusted the reference to the MPR methodology. This resulted in lines 15-36 being added to the start of the introduction.

*Lakes and reservoirs*

We reported the presence of lakes and reservoirs (lines 136-138). In 25 of the 567 basins, lakes and or reservoirs were included in the model parameters given a threshold area of 1 km2 and 10 km2 respectively. Due to the small amount this did not alter the conclusions of the study.

*Basin selection*

We clarified the reasons behind failed runs (lines 100 – 103). Besides missing streamflow observations errors occurred due to parameter estimation errors that in some cases became clear during runtime. Therefore we now only state parameter estimation error as the reason. These occur during drainage network delineation, either when the basin outlet consisted of a single grid cell that results in a model coding error or when inconsistencies occurred in the local drainage direction layer.

*Calibration methodology*

The calibration routine is now more extensively described in lines 164 - 178**.** We manually calibrated the parameter based on an interval ranging between 1 – 1000 KsatHorFrac values. The best performing model run based on streamflow estimates at the outlet is evaluated using the modified KGE score. This model instance is subsequently used for the evaluation period model run.

We have added the description as to why we calibrate the parameter KsatHorFrac in lines **164 – 171** in section 2.2.2 Model Runs & Calibration.

*Objective function relevance (KGE 0.22)*

We have added the sampling uncertainty method of Clark et al. (2021). This is included in the methods (lines 201-207), the results (lines 314-320), and the discussion (lines 419-416). The results altered the main conclusion based on the testing of the statistical difference in objective function distributions. The difference in results is now considered to be too small in the context of sampling uncertainty. We believe that this great suggestion from the reviewer helped with the discussion of the results based on objective functions.

*Forcing and model resolutions*

The effect of native forcing resolution is discussed in more depth in (lines 447 – 457) and reflects on the findings in literature. We included the need for testing this effect further in future research (outlook section).

*Scaling in hydrology*

We agree with the reviewer that spatial scaling encompasses more than only varying spatial resolution in hydrological models. Therefore we adjusted the use of the term scaling in the publication and now use varying spatial resolution instead. This is reflected in the title of the title of the publication and further discussed in the discussion and outlook sections.

*Model selection*

The reason for selecting the wflow_sbm model is now more clearly defined in the introduction. In addition, we added the benefits of using alternative delineation methods (vector-based) to the discussion section.

**Overview of Changes:**

1. Title:

    - Better describes content of the study and uses the term varying spatial resolution instead of scaling.

2. Abstract:

    - The abstract is updated based on the extra analyses we performed.

3. Introduction:

    - Restructured.

    - Includes competing basin discretization approaches.

- Includes a larger body of cited literature that refers to the land surface community, parameter identifiability problem, parameter transferability, and the representative elementary watershed.

- Clearly stated that we are not using the MPR method.

- We now use "the effects of varying spatial resolution" instead of scaling as this is only a part of scaling.

4. Methodology:

- Better description as to why basins are excluded from the analyses.

- Better description of the calibration methodology.

- Description of the sampling uncertainty of the KGE score method of Clark et al. 2021.

5. Results:

- Restructured.

- Included calibration period CDF of the KGE score and the decomposed components.

- Added the evaluation period KGE score components to the CDF figure.

- Added objective function uncertainty section. Includes the analyses similar to Clark et al. 2021. Added a table summarizing these results based on the evaluation CDF.

6. Discussion:

- Restructured. Added headers for readability.

- Refer to other large domain studies.

- Discuss the sampling uncertainty results and what this means for the statistical KS-test results.

- Discuss the effect of using coarse meteorological forcing products.

- Discuss how vector-based discretization has major benefits when it comes to computational cost and topographic discretization.

7. Outlook:

- Expended on what needs to be added to this study in order to do a complete scaling assessment as opposed to only a spatial scaling assessment.

8. Conclusions:

- State that the sampling uncertainty is large and therefore the conclusion based on the KS-test are inconclusive.

- Added "at the outlet" to the conclusion regarding finer spatial resolution does not always leads to better streamflow estimates.